# TextResNet: Decoupling and Routing Optimization Signals in Compound AI Systems via Deep Residual Tuning

**Suizhi Huang** [1 2]  **Mei Li** [1]  **Han Yu** [1]  **Xiaoxiao Li** [3 4]

## Abstract

Textual Gradient-style optimizers (TextGrad) (Yuksekgonul et al., 2025) enable gradient-like feedback propagation through compound AI systems. However, they do not work well for deep chains. The root cause of this limitation stems from the *Semantic Entanglement* problem in these extended workflows. In standard textual backpropagation, feedback signals mix local critiques with upstream contexts, leading to *Attribution Ambiguity*. To address this challenge, we propose TEXTRESNET, a framework that reformulates the optimization process to achieve precise signal routing via four key innovations. Firstly, in the forward pass, it enforces Additive Semantic Deltas to preserve an Identity Highway for gradient flow. Secondly, in the backward pass, it introduces Semantic Gradient Decomposition via a Semantic Projector to disentangle feedback into causally independent subspaces. Thirdly, it implements Causal Routing, which routes projected signals to their specific components. Finally, it performs Density-Aware Optimization Scheduling to leverage the disentangled signals to dynamically allocate resources to key system bottlenecks. Our results show that TEXTRESNET not only achieves superior performance compared to TextGrad, but also exhibits remarkable stability for agentic tasks in compound AI systems where baselines collapse. Code is available at https://github.com/JeanDiable/TextResNet.

[1]Nanyang Technological University, Singapore [2]Jinan-NTU Green Technology Research Institute, China [3]The University of British Columbia, Canada [4]Vector Institute, Canada. Correspondence to: Xiaoxiao Li <xiaoxiao.li@ece.ubc.ca>, Han Yu <han.yu@ntu.edu.sg>.

*Proceedings of the 43rd International Conference on Machine Learning*, Seoul, South Korea. PMLR 306, 2026. Copyright 2026 by the author(s).

## 1. Introduction

The shift from monolithic LLMs to Compound AI Systems (CAS) enables complex, long-horizon tasks via component orchestration (Kandogan et al., 2025; Chen et al., 2024a; Yao et al., 2023a; Shinn et al., 2023; Huang et al., 2023; Yu et al., 2014; Pan et al., 2016). However, the discrete nature of these systems poses a fundamental optimization challenge (Khattab et al., 2022; Opsahl-Ong et al., 2024; Zhang et al., 2024). Unlike neural networks, CAS cannot be optimized via standard automatic differentiation, leaving it reliant on unscalable manual engineering or black-box search (Zhou et al., 2022; Guo et al., 2023; Yang et al., 2023).

A promising breakthrough is "differentiation via text" (e.g., TextGrad (Yuksekgonul et al., 2025)), which treats CAS as differentiable graphs. While effective for shallow chains, this paradigm fails in extended workflows due to **Semantic Entanglement**: feedback signals mix local critiques with noisy upstream context (Chen et al., 2023a; Peng et al., 2024; Cemri et al., 2025; Huang et al., 2024). Without disentangling these signals, the optimizer faces **Attribution Ambiguity**, leading to three distinct problems that hinder optimization (Fig. 1):

1. **Signal Blockage:** Feedback becomes too generic before it reaches the upstream component that caused the error, leaving the root cause unaddressed, often shown as general conclusion of feedback.

2. **Downstream Over-correction:** A downstream component is asked to hallucinate fixes for an error caused by upstream components, even when its own output is faithful to its input.

3. **Upstream Pollution:** Feedback from a downstream error is incorrectly concluded and sent to upstream, causing a previously correct component to change.

Fundamentally, these problems arise because standard textual backpropagation treats state transitions as unconstrained rewrites, making it difficult to tell what should be preserved and what should be changed.

To resolve this, we draw on residual learning theories like Deep Delta Learning (DDL) (Zhang et al., 2026)

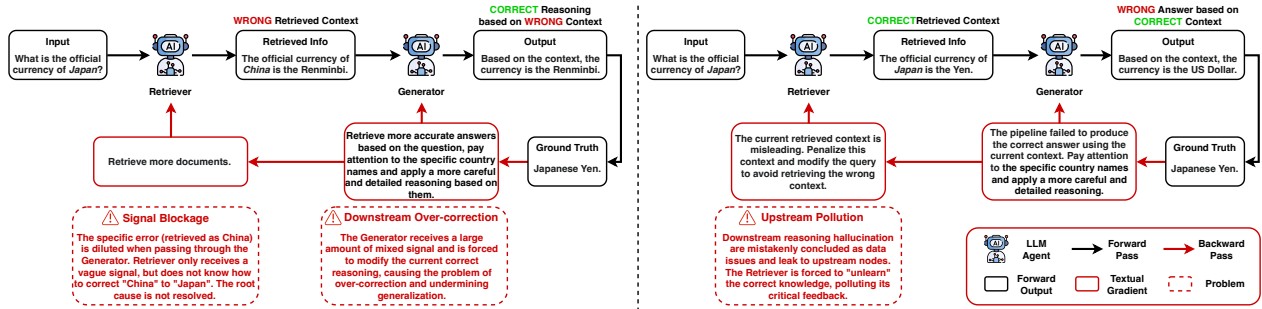

*Figure 1.* **Optimization Problems in Compound AI Systems,** an example through a simplified two-agent QA pipeline ( Info Retriever → Answer Generator). We identify three critical failure modes arising from *Attribution Ambiguity* in standard textual backpropagation. (a) **Signal Blockage**: Critical feedback fails to propagate to the upstream node. (b) **Downstream Over-correction**: Downstream nodes are forced to hallucinate fixes for upstream. (c) **Upstream Pollution**: Downstream reasoning errors mistakenly being concluded and leak to upstream nodes.

and Manifold-Constrained Hyper-Connections (mHC) (Xie et al., 2025), which address signal degradation via geometric constraints (He et al., 2016; Srivastava et al., 2015). These frameworks suggest that stable learning requires an "Identity Highway" where layers operate as controlled residual updates rather than arbitrary rewrites. We hypothesize that this "geometric disentanglement" adapts to the semantic space: by structurally separating "Identity" (upstream context) from "Delta" (local reasoning), we can resolve semantic entanglement.

Building on these insights, we introduce TEXTRESNET, a unified optimization framework designed to decouple and precisely route textual gradients in Compound AI Systems. Unlike prior approaches that treat the backward pass as a flat conversation (Yuksekgonul et al., 2025; Shinn et al., 2023), TEXTRESNET reformulates optimization as a structured routing problem via four key innovations:

- **Forward Pass via Additive Semantic Deltas:** We redefine state evolution not as lossy rewriting, but as residual refinement (Context ⊕ Delta). This enforces a structural Identity Highway that explicitly keeps historical context available while recording each component's new contribution separately, reducing Signal Blockage.

- **Backward Pass via Semantic Projector and Causal Routing:** We introduce a Semantic Projector as symbolic differentiator to decompose feedback into causally independent subspaces (Local vs. Upstream). This further enables Causal Routing: the system emits semantic STOP_GRADIENT signals for pure local failures (preventing Upstream Pollution) while allowing upstream errors to directly flow upward (preventing Downstream Over-correction).

- **Density-Aware Optimization Scheduling:** Leveraging the disentangled signals, we could track the density of strictly local errors as an unbiased estimator

of each component's contribution to the global loss. This allows the scheduler to dynamically allocate the optimization budget to the system's true bottlenecks via Boltzmann sampling.

We empirically validate TEXTRESNET on four challenging benchmarks. Our results show that TEXTRESNET not only achieves superior performance compared to TextGrad, but also exhibits remarkable stability in deep chains where baselines collapse. In-depth analysis confirms that our architecture successfully disentangles optimization signals, effectively neutralizing the effects of input noise and preventing error propagation.

Our contributions are summarized as follows:

- **Problem Identification:** We identify Semantic Entanglement and Attribution Ambiguity as root causes of TextGrad-style optimization failure in CAS, leading to three failure modes: Signal Blockage, Downstream Over-correction, and Upstream Pollution.

- **Unified Framework Design:** We propose TEXTRES-NET, which resolves these issues via four innovations: Additive Semantic Deltas, Semantic Projector, Causal Routing, and Density-Aware Scheduling.

- **Empirical Validation:** We demonstrate the effectiveness of TEXTRESNET through comprehensive experiments on four diverse benchmarks.

## 2. Related Work

**Compound AI Systems and Optimization Challenges.** The shift from monolithic LLMs to Compound AI Systems (CAS) enables complex, long-horizon tasks (Zaharia et al., 2024; Kandogan et al., 2025; Guo et al., 2025), formalized by frameworks like LangChain and DSPy (Khattab et al., 2022; Zhang et al., 2024; Yu et al., 2024). However, optimizing these modular graphs faces Attribution Ambiguity,

where downstream failures often result from compounded upstream retrieval or reasoning errors (Cemri et al., 2025; Huang et al., 2024; Chen et al., 2023a). Existing methods, including manual engineering and black-box search (Zhou et al., 2022; Yang et al., 2023; Liashchynskyi, 2019; Guo et al., 2023), lack the structural awareness to assign credit accurately, scaling poorly with complexity and failing to isolate root causes (Chen et al., 2024b).

**Textual Differentiation.** TextGrad (Yuksekgonul et al., 2025) pioneered "differentiation via text," treating CAS as differentiable graphs. However, by modeling forward passes as lossy rewriting and backward passes as flat conversations (Shinn et al., 2023; Madaan et al., 2023; Liu et al., 2023), this paradigm suffers from Semantic Entanglement, leading to the three problems identified above. To address optimization in compound systems, OPTIMAS (Wu et al., 2025) decomposes objectives via Local Reward Functions (Chen et al., 2025b; Lin et al., 2023). However, this approach relies on Reinforcement Learning (RL) with dense component-level supervision, incurring high costs for data annotation and auxiliary reward modeling (Hu et al., 2024; Chen et al., 2025c). In contrast, TEXTRESNET proposes a training-free architectural solution that enforces precise signal routing. Notably, our framework is orthogonal to OPTIMAS: while OPTIMAS relies on RL for policy updates, it lacks an explicit mechanism for causal error attribution, which TEXTRESNET structurally enforces.

**Residual Learning and Geometric Constraints.** Deep learning addresses signal degradation via ResNet's identity skip connections (He et al., 2016). Recent theories, including Deep Delta Learning (DDL) (Zhang et al., 2026) and Manifold-Constrained Hyper-Connections (mHC) (Xie et al., 2025), formalize residual learning through dynamic gating and manifold constraints to control state evolution (Liu et al., 2018; Chen et al., 2023b). We rigorously map these geometric principles to discrete textual optimization. By redefining the forward pass as a "semantic residual" (Identity + Delta) and implementing Semantic Gradient Decomposition, TEXTRESNET semantically disentangles local defects from upstream context, bringing the stability and depth-scalability of residual networks to CAS.

# 3. Preliminaries and Problem Formulation

In this section, we formalize Compound AI Systems within the framework of Stochastic Computation Graphs (SCGs) (Schulman et al., 2015). We then characterize the limitations of standard textual backpropagation, identifying Structural Ambiguity as the root cause of the problems described in Section 1. Finally, drawing from residual learning theory, we establish the necessary geometric design principles required to resolve these problems.

## 3.1. Compound AI Systems as Stochastic Computation Graphs

We represent a Compound AI System (CAS) as a directed acyclic graph $\mathcal{G} = (\mathcal{V}, \mathcal{E})$. Nodes $v \in \mathcal{V}$ represent computational modules (e.g., LLM agents, tools), and edges $(u, v) \in \mathcal{E}$ represent the flow of semantic information.

**Forward Pass:** Let the nodes be topologically ordered as $v_1, \ldots, v_L$. Each node $v_l$ maintains a learnable configuration $\theta_l$ (e.g., system prompts) and produces a hidden state $h_l \in \mathcal{H}$ (output text). In standard agentic workflows, the state transition is governed by a stochastic operator $\mathcal{M}_l$:

$$h_l \sim P(h_l \mid h_{l-1}, \theta_l). \tag{1}$$

Crucially, standard LLM agents typically model $\mathcal{M}_l$ as an unconstrained rewriting process (Yao et al., 2023b; Shinn et al., 2023). Unlike Residual Networks where $h_l = h_{l-1} + f(h_{l-1})$ (He et al., 2016), an LLM agent consumes the input $h_{l-1}$ and generates a completely new sequence $h_l$. This creates a **lossy state transition**, where the identity of the input context is implicitly merged with the transformation logic. Consequently, the final output $y = h_L$ is a composite function where the contribution of $\theta_l$ is coupled with the propagated features of $h_{l-1}$.

**Optimization Objective.** Given a task distribution $\mathcal{D}$ and a global reward function $R(y, y^*)$, we seek to maximize:

$$\Theta^* = \arg\max_{\Theta} \mathbb{E}_{x \sim \mathcal{D}, h \sim P_\Theta(\cdot \mid x)}[R(h_L, y^*)]. \tag{2}$$

## 3.2. The Problem: Structural Ambiguity

Recent frameworks approximate gradients via natural language feedback (Yuksekgonul et al., 2025). We define a Textual Gradient $g_l$ as a semantic critique generated by a "Backward LLM" operator $\mathcal{B}$: $g_l = \mathcal{B}(g_{l+1}, h_l, h_{l-1}, \theta_{l+1})$.

In differentiable programming, the chain rule relies on the independence of partial derivatives (local vs. transport Jacobians). However, in the discrete textual space, standard backpropagation treats the backward pass as a flat conversation. This leads to a fundamental credit assignment problem:

**Definition 3.1** (Semantic Attribution Ambiguity). Let $\mathcal{E}_{\text{local}}$ and $\mathcal{E}_{\text{upstream}}$ be the sets of true error sources originating from the local parameter $\theta_l$ and the input state $h_{l-1}$, respectively. Structural Ambiguity occurs when the backward operator $\mathcal{B}$ generates a feedback signal $g_l$ that mixes these sources, such that the optimizer cannot determine if the correction should target the logic ($\theta_l$) or the context ($h_{l-1}$).

This ambiguity leads to three problems: Signal Blockage occurs when $g_l$ fails to capture errors in $\mathcal{E}_{\text{upstream}}$; Downstream Over-correction occurs when $g_l$ falsely attributes upstream errors to $\theta_l$; and Upstream Pollution occurs when $g_l$ falsely propagates local reasoning errors to $h_{l-1}$.

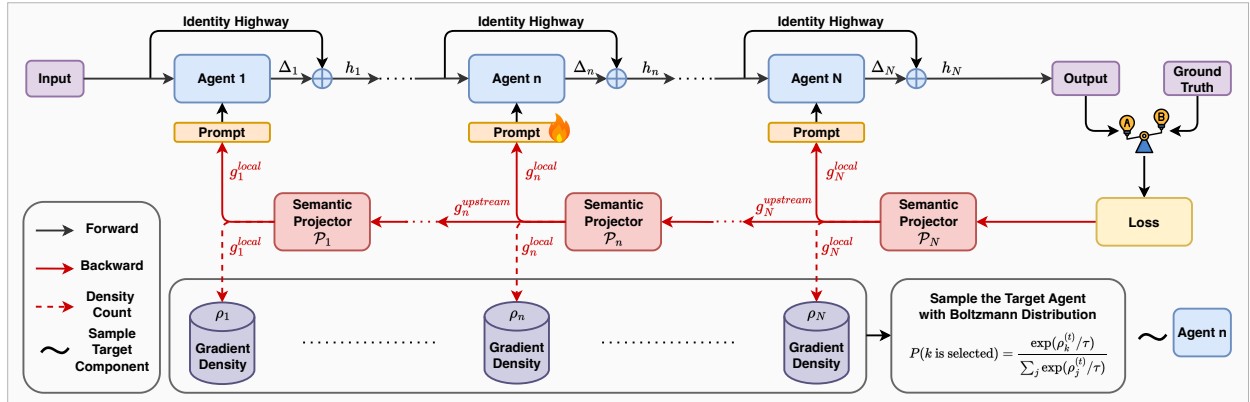

*Figure 2.* **Overview of the TEXTRESNET Framework.** Our approach reformulates optimization as a structured semantic routing problem across three stages: (1) **Forward Pass**: Agents generate Additive Semantic Deltas ($\Delta$) rather than rewrites, establishing an Identity Highway that preserves upstream context for attribution. (2) **Backward Pass**: The Semantic Projector ($\mathcal{P}$) enforces Semantic Gradient Decomposition, projecting feedback into causally independent subspaces ($g^{\text{local}}$, $g^{\text{upstream}}$) to implement precise Causal Routing. (3) **Optimization**: A Density-Aware Scheduler tracks the accumulation of local errors (Gradient Density $\rho$) to dynamically allocate the optimization budget to true system bottlenecks via Boltzmann sampling.

## 3.3. Geometric Principles for Stable Optimization

To resolve Structural Ambiguity, we look to the geometric principles of Deep Delta Learning (DDL) (Zhang et al., 2026) and Manifold-Constrained Hyper-Connections (mHC) (Xie et al., 2025). We map these theoretical insights to the semantic space to define two core design principles for our framework:

**Definition 3.2** (Design Principle 1: Lossless Context Preservation). To prevent Signal Blockage, the forward mapping must satisfy the Reconstructible Property. Instead of lossy rewriting, the state evolution should be modeled as an additive refinement:

$$h_l = h_{l-1} \oplus \Delta_l. \tag{3}$$

This ensures that the upstream context $h_{l-1}$ remains explicitly accessible in $h_l$, guaranteeing that the "Identity" of the input is preserved for gradient attribution at any depth $L$ (Xie et al., 2025).

**Definition 3.3** (Design Principle 2: Semantic Disentanglement). To prevent Downstream Over-correction and Upstream Pollution, the optimization signal must be structurally disentangled. We require the feedback $g_l$ to be projected into two causally independent subspaces, $\mathcal{S}_{\text{local}}$ and $\mathcal{S}_{\text{upstream}}$. This condition implies that an update to the local mechanism $\theta_l$ derived from the local feedback component imposes no constraints on the validity of the upstream context $h_{l-1}$, thereby isolating local defects from upstream defects. (Zhang et al., 2026).

These two principles form the basis for TEXTRESNET: Principle 1 leads to a residual forward architecture, while Principle 2 leads to a projective backward operator.

## 4. Methodology: The TEXTRESNET Framework

We introduce TEXTRESNET, a unified optimization framework designed to decouple and precisely route textual gradients in Compound AI Systems. Our approach reformulates the optimization process from a flat textual conversation into a structured semantic routing problem.

### 4.1. Overview of TEXTRESNET

The workflow of TEXTRESNET operates in three synchronized stages, as illustrated in Fig. 2.

First, in the **Forward Pass**, we prevent Signal Blockage by enforcing an **Identity Highway**. Each component acts as a learner of semantic deltas (increments) rather than performing full state rewrites. This explicitly separates the historical context from the local transformation.

Second, in the **Backward Pass**, we introduce the **Semantic Projector** as symbolic differentiator. Instead of propagating mixed feedback, this operator dynamically projects incoming error signals onto disentangled semantic subspaces (Local vs. Upstream). This allows the system to implement **Causal Routing**: specifically, when an error is identified as purely local, the system emits a semantic STOP_GRADIENT signal to the upstream node. This explicit stop signal prevents Upstream Pollution by confirming that the upstream output was valid. Conversely, pure upstream errors flow directly along the identity highway to prevent Downstream Over-correction.

Finally, an **Optimization Scheduler** leverages these disentangled signals. By tracking the density of strictly local

errors, the scheduler could gain an unbiased estimate of each component's true contribution to the global loss, allowing for efficient allocation of optimization resources via Boltzmann sampling.

### 4.2. Forward Pass: Additive Semantic Deltas

To support semantic decomposition, the forward pass must structurally separate the "identity" (context) from the "transformation" (local reasoning).

**The Additive Operator.** We define the output state $h_l$ of node $v_l$ as the semantic summation of its preserved input context and a generated residual:

$$h_l = h_{l-1} \oplus \mathcal{F}(h_{l-1}; \theta_l), \tag{4}$$

where $\mathcal{F}(\cdot; \theta_l)$ represents the component's specific contribution (the Semantic Delta $\Delta_l$), and $\oplus$ denotes a semantic addition operator. In the discrete textual domain, $\oplus$ is instantiated as structured concatenation or patch application (e.g., appending a reasoning step or applying a code diff), ensuring that the information in $h_{l-1}$ remains explicitly accessible in $h_l$ (see Appendix B.5 for implementation detail).

**Proposition 4.1** (Information Preservation in Residual Chains). *Let $\mathcal{H}$ be the semantic state space. In a deep chain of length $L$ employing additive semantic deltas, the upstream context $h_0$ remains theoretically accessible in the final state $h_L$. Unlike lossy rewriting chains where information decays exponentially with depth, our additive structure ensures that upstream context is available for causal attribution during the backward pass (detailed in Appendix D).*

**Context-Window Management.** While TEXTRESNET encourages an "identity highway" across depth, our implementation does not rely on monotonic concatenation of all intermediate texts. Instead, the forward execution maintains a shared *dict context* that accumulates component outputs by key-wise update, and each component only receives the *minimal required slice* specified by its input_fields. Thus, the identity highway should be understood as a structured state-preservation constraint implemented through key-wise memory and deterministic field selection, rather than as unlimited full-context concatenation. Concretely, at depth $\ell$, the component call is formed from $x_\ell := \Pi_\ell(h_\ell)$ where $\Pi_\ell$ is the interface-level projection (field selection) induced by input_fields; this prevents most components from ever seeing the full trajectory. More details are provided in Appendix B.6.

### 4.3. Backward Pass: Semantic Projector and Causal Routing

To resolve Structural Ambiguity, we implement a **Semantic Projector** $\mathcal{P}$ that enforces Semantic Gradient Decomposition as the symbolic differentiator.

**Semantic Gradient Decomposition.** We assume that the entangled gradient $g_l$ can be projected into two causally independent components: $g^{\text{local}}$ (actionable by $\theta_l$) and $g^{\text{upstream}}$ (actionable by $v_{l-1}$). The projector $\mathcal{P}_l$ performs this decomposition as follows:

$$g_l^{\text{local}} = \mathcal{P}_l^{\text{local}}(g_l), \tag{5}$$
$$g_l^{\text{upstream}} = \mathcal{P}_l^{\text{upstream}}(g_l). \tag{6}$$

In practice, each $\mathcal{P}_l$ is parameterized by a Backward LLM instructed to perform causal attribution. Based on this decomposition, we implement a **Causal Routing**:

**Pure Local Defect:** If the Projector determines the error is specific to the current component, it emits a semantic STOP_GRADIENT signal to the upstream node, explicitly informing it that its output is valid.

**Pure Upstream Defect:** If the Projector determines the error stems from invalid input, it sets $g_l^{\text{local}} = \emptyset$. The signal flows directly along the Identity Highway ($g_l^{\text{upstream}} \neq \emptyset$) without triggering local updates, protecting the current component from overfitting to upstream noise.

**Mixed Fault:** If both subspaces contribute to the error, the signal is split. $g_l^{\text{local}}$ updates $\theta_l$, while $g_l^{\text{upstream}}$ will propagates to upstream $v_{l-1}$.

To illustrate using a RAG task with two agents: a Retriever followed by a Generator. The Causal Routing of the Generator has three possibilities: If the retrieved context of the Retriever is correct but its generated answer is wrong, it is a Pure Local Defect. In contrast, if the retrieved context is wrong and the answer follows it, it is a Pure Upstream Defect. For the third case, if the context is partial and the answer contains hallucinations, it is a Mixed Fault.

**Proposition 4.2** (Bounded Error Propagation Analysis). *Let $SNR(g_l)$ be the ratio of actionable signal variance to non-actionable noise variance at depth $l$. We model the error accumulation in the semantic space as a linear additive process with variance $\sigma^2$. Under this simplified model, if there exists a non-zero probability $p$ that a noise component is identified as local-specific and filtered out, the Causal Routing mechanism ensures that the noise variance in $SNR(g_l^{TextResNet})$ is bounded by a constant $\frac{1-p}{p}\sigma^2$ independent of depth $L$. In contrast, without routing, the noise variance grows linearly with $L$.*

This analysis (detailed in Appendix D) suggests that our filtering strategy effectively mitigates the "Upstream Pollution" problem by truncating irrelevant feedback chains, maintaining stable optimization signals even in deep chains.

### 4.4. Optimization Scheduler: Density-Aware Scheduling

With signals routed to their likely sources, the volume of local feedback accumulating at a node provides a practical

*Table 1.* Performance comparison across compound AI benchmarks. Best in bold and second best with underline.

| Method | HotpotQA (F1) | BigCodeBench (Pass%) | PubMedQA (Acc) | STARK-PRIME (MRR) |
|---|---|---|---|---|
| CoT | $33.92 \pm 0.76$ | $34.15 \pm 1.43$ | $57.34 \pm 1.12$ | $39.76 \pm 0.84$ |
| HBC | $21.16 \pm 0.97$ | $27.78 \pm 2.08$ | $58.80 \pm 0.58$ | $36.95 \pm 0.59$ |
| DSPy | $\underline{44.90 \pm 0.32}$ | $33.81 \pm 2.75$ | $\underline{60.26 \pm 0.40}$ | $\underline{41.40 \pm 0.04}$ |
| TextGrad | $24.86 \pm 1.19$ | $\underline{35.71 \pm 0.10}$ | $56.96 \pm 2.24$ | $41.31 \pm 1.67$ |
| TextGrad (w/ Sum) | $24.12 \pm 1.25$ | $35.12 \pm 0.67$ | $56.12 \pm 1.85$ | $40.72 \pm 1.21$ |
| **TEXTRESNET (Ours)** | $\mathbf{46.23 \pm 1.15}$ | $\mathbf{37.86 \pm 0.45}$ | $\mathbf{60.31 \pm 1.51}$ | $\mathbf{41.75 \pm 0.85}$ |

proxy for its contribution to the global error. We define the Gradient Density $\rho_i^{(t)}$ for node $v_i$ as the accumulated count of locally-projected feedback messages ($g_i^{\text{local}}$) since the last update. Formally, $\rho_i^{(t)} = \sum_{\tau=t_{\text{last}}}^{t} \mathbb{I}(g_{i,\tau} \in \mathcal{S}_{\text{local}})$, where $t_{\text{last}}$ is the step of the last update. We then allocate the optimization budget using a Boltzmann distribution to choose the component to optimize at each step:

$$P(k \text{ is selected}) = \frac{\exp(\rho_k^{(t)}/\tau)}{\sum_j \exp(\rho_j^{(t)}/\tau)}, \qquad (7)$$

where temperature parameter $\tau$ regulates the entropy of the scheduling policy, controlling the trade-off between focusing strictly on the worst-performing bottlenecks and maintaining gradients for other components.

This policy dynamically focuses compute on the system's true bottlenecks, implementing a robust form of coordinate descent (Wright, 2015). Unlike random scheduling which wastes resources on stable components, our density-aware approach accelerates convergence by prioritizing nodes with high local error rates (Falkner et al., 2018).

## 5. Experiments

We empirically evaluate TEXTRESNET on four challenging Compound AI System benchmarks that require multi-step reasoning. Our experimental design aims to validate the framework's effectiveness in resolving Attribution Ambiguity and ensuring precise signal routing in deep workflows.

### 5.1. Experimental Setup

**Datasets and Benchmarks.** We evaluate TEXTRESNET on four diverse multi-step reasoning and tool-orchestration benchmarks, which are proven diverse and used (Wu et al., 2025). **HotpotQA** (Yang et al., 2018) targets multi-hop RAG (full-wiki, distractor) using a 5-component pipeline (Rewriter, Extractor, Retriever, Hinter, Answerer) with 1000/250/100 train/dev/test splits. **BigCodeBench** (Zhuo et al.) assesses complex code generation via a 4-component iterative refinement pipeline; we use a subset of the full-instruction split (500/25/70 tasks). **PubMedQA** (Jin et al., 2019) tests biomedical reasoning (yes/no/maybe) using the

*Table 2.* **Step-wise Ablation Study.** We progressively add components to the TextGrad baseline to evaluate the performance gain. Res: Additive Semantic Deltas; Proj: Semantic Gradient Decomposition; Route: Causal Routing; Sched: Density-Aware Scheduling. Best in bold, second best underlined.

| Res | Proj | Route | Sched | HotpotQA (F1) | BigCode (Pass%) |
|---|---|---|---|---|---|
| ✗ | ✗ | ✗ | ✗ | 24.86 | 35.71 |
| ✓ | ✗ | ✗ | ✗ | 32.15 | 36.07 |
| ✓ | ✓ | ✗ | ✗ | 36.69 | 36.50 |
| ✓ | ✓ | ✓ | ✗ | $\underline{37.71}$ | $\underline{36.70}$ |
| ✓ | ✓ | ✓ | ✓ | **46.23** | **37.86** |

original "expert" split (475/25/500). **STARK-PRIME** (Wu et al., 2024) focuses on semi-structured retrieval and query optimization (495/51/96 queries).

**Baselines.** We compare TEXTRESNET against a diverse set of prompt optimization methods. **Chain-of-Thought (CoT)** (Wei et al., 2022) serves as the unoptimized lower bound. **HBC** (Le et al., 2018) is a hierarchical imitation learning method adapted for component-level supervision. **DSPy** (Khattab et al., 2022) represents the state-of-the-art in programmatic prompt compilation using the MIPRO optimizer (Opsahl-Ong et al., 2024). **TextGrad** (Yuksekgonul et al., 2025) is the primary baseline for textual backpropagation. To ensure a rigorous comparison, we also include **TextGrad+Sum** (Chen et al., 2025a), an enhanced variant using summarization to compress feedback, addressing context overflow issues.

**Implementation Details.** Across all benchmarks, we adopt the SCG configurations of OPTIMAS (Wu et al., 2025) to ensure topological consistency. However, we exclude OPTIMAS from direct comparison due to the fundamental divergence in supervision and training requirements discussed in Section 2. While TEXTRESNET operates as a training-free optimizer that focus on precise signal attribution, OPTIMAS relies on constructing Local Reward Functions and performing RL-based training (often involving parameter fine-tuning). This heavy dependence on auxiliary reward modeling and expensive local annotations makes the direct comparison with our gradient routing approach unfair. For detailed experimental setups, please refer to Appendix B.

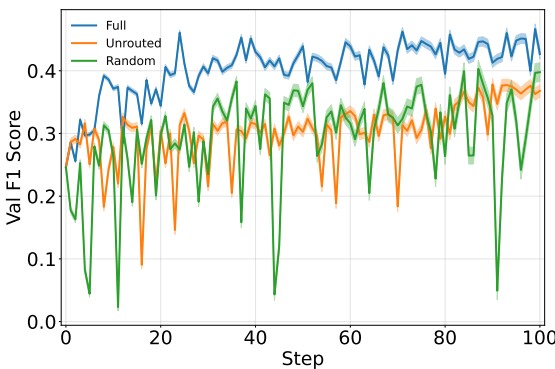

*Figure 3.* **Optimization Trajectories.** The unrouted variant (Orange) removes Causal Routing but retains Density-Aware Scheduling; it stabilizes slowly due to noise accumulation. While the random variant (Green) keeps Causal Routing but removes Density-Aware Scheduling, it suffers from high variance. TEXTRESNET (Blue) achieves stable and fast convergence.

## 5.2. Main Results

Table 1 summarizes the performance across all benchmarks. TEXTRESNET achieves consistent improvements over all baselines, with particularly strong gains on complex reasoning benchmarks.

Specifically, on the deep reasoning task HotpotQA, TEX-TRESNET surpasses the original TextGrad by a large margin (+21.37 F1). This gap suggests that TextGrad struggles with the 5-component pipeline's depth. Instead, our TEX-TRESNET effectively disentangles these signals, preserving the optimization direction and achieves better performance. Similarly, on BigCodeBench, where precise code corrections are critical along the chain, TEXTRESNET achieves a 37.86% pass rate, improving upon DSPy (33.81%) and TextGrad (35.71%). Notably, the TextGrad+Sum baseline often underperforms the vanilla TextGrad. This suggests that naive summarization leads to even worse semantic entanglement, whereas our residual decomposition preserves the full semantic entropy of the error signal and routes them to the correct components.

## 6. Ablation Studies and In-depth Analysis

In this section, we break down TEXTRESNET to investigate the internal mechanisms driving its performance. We aim to empirically verify two core hypotheses: (1) that the architectural components (Forward Residuals, Backward Projection, Causal Routing, and Density-Aware Scheduling) contribute collectively to performance; and (2) that the system correctly disentangles causal errors from noise and routes them to the correct components, effectively resolving the problems of Attribution Ambiguity. We conduct these analyses primarily on HotpotQA and BigCodeBench, as their long-horizon dependency chains provide a rigorous testbed for signal propagation stability. More in-depth analysis is pro-

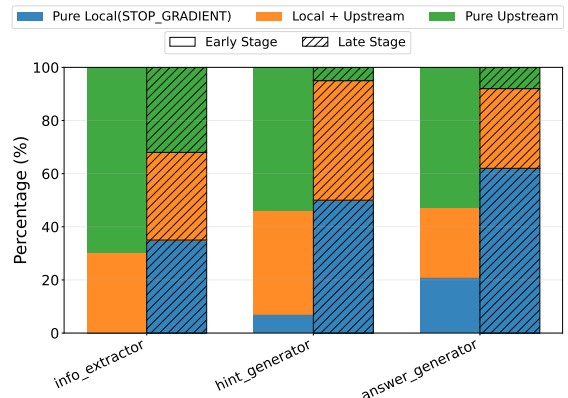

*Figure 4.* **Evolution of Error Attribution.** We focus on the three downstream learnable components (InfoExtractor, HintGenerator, AnswerGenerator). It could be observed that the system transfer from propagating errors upstream (Early Stage) to resolving them locally (Late Stage), resembling curriculum learning process.

vided in Appendix F. In particular, Appendix F.5 clarifies that chain depth is best viewed as a stress test for long attribution paths rather than the only driver of performance gain; the largest improvements appear when depth coincides with strong inter-component dependency. Appendix F.7 further provides a failure-mode annotation study and a side-by-side example showing how routed feedback resolves a concrete signal-blockage case.

### 6.1. Ablation Study

To rigorously quantify the contribution of each module, we conduct a progressive ablation study starting from the TextGrad baseline. We incrementally incorporate our proposed components: (1) Additive Semantic Deltas in the forward pass; (2) Semantic Gradient Decomposition via the Semantic Projector in the backward pass; (3) Causal Routing via semantic filtering; and (4) Density-Aware Scheduling.

Table 2 summarizes the results, revealing a clear monotonic improvement. The incorporation of Additive Semantic Deltas alone yields a significant gain (+7.29 F1 on Hot-potQA), confirming that the structural "Identity Highway" effectively mitigates information loss during forward propagation. Introducing Semantic Gradient Decomposition could provide significant gains in isolation (+4.54 F1), and its combination with Causal Routing further triggers a performance jump. Finally, the Density-Aware Scheduling further boosts performance (+8.52 F1), suggesting that allocating computational budget to nodes with high "Gradient Density" (true bottlenecks) is far more efficient than uniform or random scheduling, resulting in better performance with the same training steps.

**Optimization Dynamics.** Figure 3 visualizes the training trajectories of our TEXTRESNET and its two ablation variants. The Random Schedule variant (Green) eliminates the

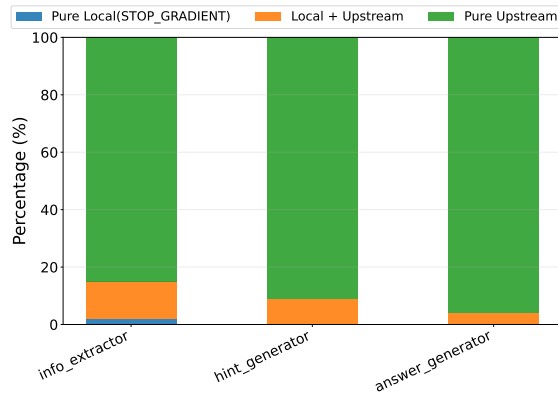

*Figure 5.* **Attribution Accuracy under Batch Shuffling.** We focus on the three downstream learnable components (InfoExtractor, HintGenerator, AnswerGenerator). We measure how often the optimizer correctly attributes the error to the shuffled input (Upstream) versus attempting to fix the prompt (Local).

Density-Aware Scheduling from the full TEXTRESNET, it suffers from high variance and oscillation, indicating that the random scheduler is not chasing the real bottleneck component, but those without necessary modification needs. The Unrouted variant (Orange) eliminates the Causal Routing from the full TEXTRESNET, it stabilizes the process but converges slowly. In contrast, the full TEXTRESNET (Blue) demonstrates a steep convergence curve, rapidly identifying and resolving bottlenecks to reach a higher optimum.

### 6.2. System Dynamics: The Phase Transition of Error Routing

Deep Delta Learning theory proposes that an optimal system should dynamically adjust its signal routing based on the training stage. To verify this, we track the attribution of the Semantic Projector throughout the training process on HotpotQA. Specifically, we calculate the proportion of feedback signals projected into the Local Subspace ($\mathcal{S}_{\text{local}}$) versus the Upstream Subspace ($\mathcal{S}_{\text{upstream}}$) at each epoch.

As shown in Figure 4, we observe a distinct **Phase Transition** in error routing:

**Early Stage (0-10 Steps):** The system is dominated by upstream signal propagation. Downstream nodes (e.g., AnswerGenerator) correctly identify that their failures stem from poor upstream context rather than internal logic. By routing gradients primarily to $\mathcal{S}_{\text{upstream}}$, the system effectively resolves **Signal Blockage**, ensuring the root cause is addressed. Simultaneously, by suppressing local updates, it prevents **Downstream Over-correction**, protecting the reasoning module from overfitting to hallucinated context.

**Late Stage (Last 10 Steps):** As upstream modules stabilize, the error distribution shifts downstream. The AnswerGenerator begins to resolve reasoning issues locally. The Projector shifts its focus to $\mathcal{S}_{\text{local}}$, engaging Causal Routing to polish

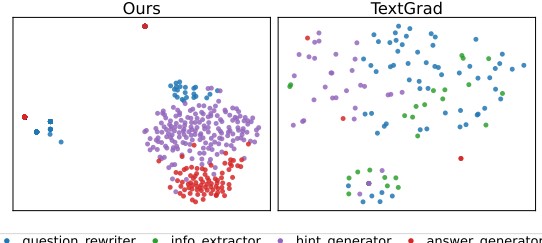

*Figure 6.* **Semantic Distribution of Gradients.** TEXTRESNET (Left) produces structurally distinct feedback clusters, whereas TextGrad (Right) generates entangled, overlapping critiques.

the prompt. Crucially, this prevents **Upstream Pollution**: reasoning errors are contained locally and are less likely to leak backward to destabilize the converged upstream components.

This dynamic shift confirms that TEXTRESNET implicitly learns a curriculum: structurally prioritizing the upstream components before optimizing the downstream components.

### 6.3. Robustness Check: Causal Attribution under Batch Shuffling

A critical property of TEXTRESNET is the ability to correctly distinguish between local errors and upstream input noise. To rigorously test this, we designed a counterfactual **Batch Shuffling Intervention** on HotpotQA.

During the forward pass, we intentionally shuffled the outputs of the first node (QuestionRewriter) before passing them to the second node (InfoExtractor). This creates a specific mismatch: the input to the InfoExtractor is valid text but semantically irrelevant to the ground truth, simulating a severe upstream hallucination. We then measured the attribution accuracy of all downstream components during the backward process.

As reported in Fig. 5, TEXTRESNET correctly identifies the mismatch as an Upstream error in 96% of cases on the last node (AnswerGenerator). By routing the signal purely to $\mathcal{S}_{\text{upstream}}$ and blocking local updates, our architecture successfully decouples the optimization process from input-dependent noise, simultaneously resolving **Signal Blockage** by notifying the upstream node and preventing **Downstream Over-correction** by freezing the local prompt.

### 6.4. Semantic Coherence via t-SNE

To visually confirm that our Semantic Gradient Decomposition effectively disentangles feedback, we collected the textual gradients generated for all components during training and projected their Sentence-BERT embeddings into a 2D space using t-SNE as illustrated in Figure 6.

The TEXTRESNET distribution (Left) demonstrates that feedback samples form distinct, well-separated clusters cor-

responding to their source components. This implies that the optimizer receives clear, component-specific instructions (e.g., "Fix retrieval keywords" vs. "Fix reasoning logic"). Conversely, the **TextGrad** distribution (Right) suffers from Semantic Entanglement where the clusters overlap significantly, indicating that upstream components are flooded with entangled feedback mixed with downstream failures.

## 7. Conclusion

We presented TEXTRESNET, a unified framework that resolves the critical optimization bottlenecks in Compound AI Systems. We identify Semantic Entanglement as a root cause of this bottleneck, where Attribution Ambiguity leads to three recurring failure modes. By mapping the geometric principles of residual learning to the semantic space, our approach includes Additive Semantic Deltas, Semantic Projector, Causal Routing and Density-Aware Scheduling to precisely disentangle and propagate feedback. Empirical results demonstrate that TEXTRESNET not only achieves superior performance but also ensures stability in deep chains where prior methods fail.

## Acknowledgment

Suizhi Huang is supported by Jinan-NTU Green Technology Research Institute (GreenTRI). Han Yu is supported by the Ministry of Education, Singapore, under its Academic Research Fund Tier 1 (RG101/24). Xiaoxiao Li is supported by the NSERC Discovery Grant RGPIN-2022-05316, NSERC Alliance Grant ALLRP 602633-24, Tri-Agency Canada; Canada CIFAR AI Chair Awards, and Canada Research Chair Fellowship; IITP grant, the Ministry of Science and ICT (RS-2024-00445087, RS- 2025-25464461), funded by the Korea government (MSIT).

## Impact Statement

This paper presents TEXTRESNET, a framework that improves the stability and optimization of Compound AI Systems. Our goal is to advance the field of Machine Learning by fixing common issues in multi-step workflows, such as unclear error tracking and signal blockage.

Our work directly improves system reliability. By correctly identifying where errors come from (via Causal Routing), our method prevents downstream agents from "hallucinating" fixes for problems caused by earlier steps. This stops errors from spreading and makes the system more robust, which is a key safety factor when deploying LLMs in real-world tasks.

Additionally, our experiments (Appendix F) show that TEXTRESNET is efficient, requiring significantly fewer tokens to reach convergence compared to existing methods. This reduces the computational cost and energy usage associated with developing and fine-tuning complex AI systems.

Finally, while automated optimization can carry risks if applied to harmful objectives, our framework operates using readable textual feedback. This transparency allows developers to easily inspect why a prompt was changed, ensuring better human oversight compared to black-box optimization methods.

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

## A. Detailed Related Work

Our work is situated at the intersection of Compound AI Systems, Automated Prompt Optimization, and architectural inductive biases for signal propagation. We review these areas below, identifying the specific gap in achieving precise signal routing to prevent the problematic error modes identified in Section 1.

**Compound AI Systems and Optimization Challenges.** The development of AI has evolved from monolithic Large Language Models (LLMs) to Compound AI Systems (CAS) to solve complex, long-horizon tasks (Lu et al., 2024; Kandogan et al., 2025; Li et al.). Frameworks like LangChain and DSPy (Khattab et al., 2022) formalize these systems as modular computation graphs. While these frameworks facilitate the construction of sophisticated workflows (Zhang et al., 2024; Yu et al., 2024), optimizing them presents a fundamental challenge: *Attribution Ambiguity* (Cemri et al., 2025; Huang et al., 2024). In a deep chain, a failure in the final output is often a compounding result of upstream retrieval errors and downstream reasoning flaws (Chen et al., 2023a). Existing optimization solutions, such as manual prompt engineering or brute-force search (Zhou et al., 2022; Yang et al., 2023; Liashchynskyi, 2019; Guo et al., 2023), lack the structural awareness to assign credit accurately. They typically treat the system as a black box, scaling poorly as system complexity increases and often failing to identify the root cause of failure (Chen et al., 2024b; 2023c).

**Textual Differentiation.** Gradient-based prompt optimization represents a paradigm shift. Recently, **TextGrad** (Yuksekgonul et al., 2025) pioneered "differentiation via text," treating the CAS as a differentiable graph where textual feedback propagates backward to update prompts. While effective for shallow networks, TextGrad treats the backward pass as a flat, unconstrained conversation, and the forward pass as a lossy rewriting process (Shinn et al., 2023; Madaan et al., 2023; Liu et al., 2023). We argue that this approach suffers from Semantic Entanglement, directly leading to the three problems defined in our Introduction. Specifically, the lack of an explicit identity path causes **Signal Blockage** (upstream components receive diluted feedback), while the inability to separate error sources leads to **Downstream Over-correction** (optimizing for noise) and **Upstream Pollution** (propagating downstream errors).

To address optimization in compound systems, OPTIMAS (Wu et al., 2025) proposes a decentralized approach by decomposing the global objective. It trains specific Local Reward Functions (LRFs) for each component to align local optimization with global performance, preventing the problem caused by flat conversational optimizing strategy (Chen et al., 2025b; Lin et al., 2023). While effective, these methods represent a learning-based solution, requiring the collection of preference data and the training of auxiliary reward models (Hu et al., 2024; Chen et al., 2025c). In contrast, our TEXTRESNET proposes an architectural solution. By enforcing architectural constraints rather than learning auxiliary models, we achieve precise signal routing without the overhead of training reward functions.

**Residual Learning and Geometric Constraints.** In deep learning, the challenge of training deep networks was addressed by ResNet, which introduced identity skip connections ($y = x + f(x)$) to create a "gradient highway" (He et al., 2016). Recent theoretical advancements, such as Deep Delta Learning (DDL) (Zhang et al., 2026) and Manifold-Constrained Hyper-Connections (mHC) (Xie et al., 2025), have further formalized residual learning as a mechanism for controlling state transitions. DDL demonstrates that dynamic gating between identity mapping and semantic projection is key to stable signal propagation (Zhang et al., 2026), while mHC emphasizes that constraining updates to a specific manifold prevents signal degradation across depths (Xie et al., 2025; Liu et al., 2018; Chen et al., 2023b).

Our work is the first to rigorously map these geometric principles to the discrete textual optimization space to resolve semantic entanglement. We redefine the forward pass of an agent not merely as a "rewrite" operation, but as a "semantic residual" (Identity + Delta). This structural inductive bias allows us to implement Semantic Gradient Decomposition during the backward process. By semantically disentangling "local defects" from "upstream context", we bring the stability and depth-scalability of residual networks to Compound AI Systems, ensuring that optimization signals are routed precisely to their causal components.

## B. Detailed Experimental Setup

This section provides complete experimental and implementation details for TEXTRESNET, including datasets, system pipelines, baselines, optimization hyperparameters, and *full prompt templates* for all LLM-driven components.

## B.1. Datasets

We evaluate on four tasks spanning multi-hop QA, code generation with execution-based verification, biomedical yes/no QA, and knowledge-base entity retrieval with heterogeneous evidence.

### B.1.1. HOTPOTQA (MULTI-HOP QA)

**Data source.** We use the standard HotpotQA (Yang et al., 2018) question–answer pairs. Each example contains:

- **Input**: `question`
- **Ground truth**: `gd_answer`

**Splits.** We use a predefined train/dev split for training and validation, and evaluate on a distractor-style test set when available; otherwise we fall back to the default test split.

**Metric.** We report answer F1 between predicted short answer and gold answer.

### B.1.2. BIGCODEBENCH (CODE GENERATION + UNIT TESTS)

**Data source.** We use the HuggingFace dataset `bigcode/bigcodebench` (version v0.1.3) (Zhuo et al.). Each example provides:

- **Instruction prompt** (used as `question`)
- **Unit tests** (used as `unit_tests`)
- **Task ID** and a fixed **entry point** (`task_func`)

**Splits.** We construct a randomized split mask with a fixed seed: a training pool of size `train_size` (default 1000) and the remaining examples as the test pool; the training pool is further split 95/5 into train/validation.

**Metric.** We report pass@1 as a binary score from sandboxed execution: 1.0 iff the final generated code passes the provided unit tests.

### B.1.3. PUBMEDQA (BIOMEDICAL YES/NO/MAYBE QA)

**Data source.** We use a PubMedQA-style dataset stored as JSONL (train and test) (Jin et al., 2019). Each example contains:

- **Input**: `question` and `context` (a paragraph or a list of sentences concatenated into one string).
- **Ground truth**: `groundtruth` in {yes, no, maybe}.

**Splits.** We form train/validation by a 95/5 split of the training JSONL (after optional shuffling/subsampling), and use the provided test JSONL as test.

**Metric.** We report exact-match accuracy on the normalized label (`yes`/`no`/`maybe`). The model output is post-processed by extracting the first occurrence of one of these labels.

### B.1.4. STARK (ENTITY RETRIEVAL WITH RELATIONAL + TEXTUAL EVIDENCE)

**Data source.** We use the STaRK dataset (Wu et al., 2024) from HuggingFace (`snap-stanford/stark`), Prime subset. Each query comes with gold answer entity IDs.

**Candidate construction and evidence.** For each query, we retrieve candidate entities using dense similarity (precomputed embeddings). We then form a 5-candidate set containing (i) at least one gold answer when possible and (ii) distractor candidates. For each candidate we provide:

- **Relational evidence**: structured relation strings (truncated to 1024 characters).

- **Textual evidence**: property/summary strings (truncated to 1024 characters).

- **Embedding similarity scores**: a length-5 list (one per candidate).

**Subset sizes.** To control cost, we evaluate on fixed-size subsets (Prime): train 250, validation 25, test 100.

**Metric.** We report MRR computed from the final ranked list of the 5 candidates.

### B.2. Systems (Pipelines)

We evaluate on four systems. For clarity, we describe each pipeline using the exact component names used in the system graphs. All these systems are exactly the implementation from OPTIMAS where a more detailed description could be found (Wu et al., 2025).

#### B.2.1. HOTPOTQA SYSTEM

**Components:** `question_rewriter` → `info_extractor` → `wikipedia_retriever` → `hint_generator` → `answer_generator`.

**Dataflow:** `question_rewriter` produces `rewritten_query`; `info_extractor` produces `search_keywords`; `wikipedia_retriever` returns `retrieve_content`; `hint_generator` produces `hints`; `answer_generator` outputs the final `answer`.

#### B.2.2. BIGCODEBENCH SYSTEM

**Components:** `code_generator` → `unit_test_generator` → `executor` → `final_code_generator`.

**Dataflow:** `code_generator` produces `initial_code`; `unit_test_generator` produces `additional_unit_tests`; `executor` returns `execution_result`; `final_code_generator` outputs final `code`.

#### B.2.3. PUBMEDQA SYSTEM

**Components:** `context_model_selector` → `context_analyst` → `solver_model_selector` → `problem_solver`.

**Dataflow:** `context_analyst` produces `summary` from `context+question`; `problem_solver` outputs the final `answer` in {`yes,no,maybe`}. (Selectors output a model choice used by the subsequent component.)

#### B.2.4. STARK SYSTEM

**Components:** `relation_scorer`, `text_scorer`, `final_scorer`.

**Dataflow:** Given `question` and five candidates with `relation_info`, `text_info`, and `emb_scores`: `relation_scorer` outputs `relation_scores`; `text_scorer` outputs `text_scores`; `final_scorer` aggregates them into `final_scores` used for ranking.

### B.3. Baselines

We follow the baseline suite and reproduction protocol used in OPTIMAS for the same set of compound-system benchmarks (PubMedQA, STaRK-Prime, HotpotQA, BigCodeBench). All baselines share the *same system graphs* (same component boundaries and I/O interfaces) and differ only in how they select/update the optimizable component configurations (primarily prompts).

**Unoptimized / Chain-of-Thought (CoT).** This baseline uses default component settings with *no learning and no prompt updates*. For reasoning-heavy components, we use standard step-by-step (CoT-style) prompting as the unoptimized reference, matching the conventional "no optimization" setup.

**HBC.** HBC is a hierarchical imitation learning baseline that aligns sub-module outputs to trajectories with high global reward. We implement HBC on the collected preference dataset by replacing the original reward model

*Table 3.* TEXTRESNET optimization-loop hyperparameters.

| Category | Setting |
|---|---|
| Total budget $T$ | 100 optimization steps |
| Training examples per step $K$ | 8 |
| Dev evaluation frequency (`update_freq`) | every 1 step |
| Repeated dev trials (`eval_time`) | 3 |
| Test repeats | 3 |
| Max evaluation concurrency | 20 |
| Random seed | 42 |

with an embedding-similarity reward: for each input, we embed the module output and the preferred output using `text-embedding-3-small`, compute cosine similarity, and weight the similarity by the score gap between the preferred and rejected outputs to form the learning signal.

**TextGrad.** TextGrad performs *global textual backpropagation* over the full system computation graph, propagating feedback hop-by-hop without explicit length constraints. Following OPTIMAS reproduction settings, we use GPT-4o-mini to optimize each component prompt independently in separate epochs, evaluate on a fixed held-out validation set every two optimization steps, and select the best-performing prompt. We use a batch size of 4 across components and datasets.

**TextGrad + Summarization (TextGrad w/ Sum).** This baseline applies summarization prompting to *compress stacked textual gradients* before they are propagated further upstream, thereby controlling feedback length. We cap the feedback to a small fixed budget (e.g., 100 tokens) while keeping other TextGrad settings unchanged. This baseline tests whether naive compression can resolve exploding-text issues without harming gradient utility.

**DSPy (MIPRO).** DSPy is a modular programmatic framework that learns and compiles prompts. We use DSPy's MIPRO optimizer, but for fair comparison, we disable few-shot demonstration selection and system-prompt optimization, and optimize only the user instruction text. We set the number of MIPRO iterations dynamically to match the controlled budget of system runs used for other baselines.

## B.4. Hyperparameters and Full Configuration

This section enumerates all implementation-critical hyperparameters for reproducibility, including (i) optimization-loop parameters, (ii) forward-pass inference settings, (iii) backward (Semantic Projector) settings, (iv) execution / tool settings, and (v) the prompt-optimizer settings.

We distinguish three LLM roles: forward-pass agents, the backward Semantic Projector, and the PromptOptimizer. We report model name, decoding temperature, and maximum generation length for each role.

**Forward-pass agents.** Forward-pass components use deterministic decoding by default (temperature 0) and task-dependent `max_new_tokens`.

**Backward Semantic Projector.** The backward model is invoked with a non-zero temperature (0.4) to improve robustness of causal attribution while remaining stable under output constraints.

**PromptOptimizer.** The prompt optimizer uses a separate decoding configuration with slightly higher temperature (0.7) to explore candidate prompt edits.

**BigCodeBench sandboxed execution.** We evaluate candidate code with sandboxed execution and fixed resource limits:

- max_as_limit = 300KB, max_data_limit = 300KB, max_stack_limit = 300KB;

- min_time_limit = 2s, gt_time_limit = 5s.

Trace and stderr are truncated (each to 3000 characters) before being passed to downstream components.

*Table 4.* Forward-pass LLM settings (model, temperature, and max generation length).

| System | Component | Model | Temp / Max new tokens |
|---|---|---|---|
| HotpotQA | question_rewriter | openai/gpt-4o-mini | 0.0 / 1024 |
| HotpotQA | info_extractor | openai/gpt-4o-mini | 0.0 / 1024 |
| HotpotQA | hint_generator | openai/gpt-4o-mini | 0.0 / 512 |
| HotpotQA | answer_generator | openai/gpt-4o-mini | 0.0 / 256 |
| BigCodeBench | code_generator | anthropic/claude-3-haiku | 0.0 / 2048 |
| BigCodeBench | unit_test_generator | anthropic/claude-3-haiku | 0.0 / 1024 |
| BigCodeBench | final_code_generator | anthropic/claude-3-haiku | 0.0 / 2048 |
| PubMedQA | context_analyst | (selected from candidates; default gpt-4o-mini) | 0.0 / 4096 |
| PubMedQA | problem_solver | (selected from candidates; default gpt-4o-mini) | 0.0 / 4096 |
| STaRK | relation_scorer | anthropic/claude-3-haiku-20240307 | 0.6 / 256 |
| STaRK | text_scorer | anthropic/claude-3-haiku-20240307 | 0.6 / 256 |

*Table 5.* Backward Semantic Projector settings.

| Parameter | Setting |
|---|---|
| Model | openai/gpt-4o-mini |
| Temperature | 0.4 |

**HotpotQA Wikipedia retrieval.** We retrieve a bounded number of snippets with top-$k$ set to 20.

**STaRK evidence truncation.** Per-candidate relational evidence and textual evidence are each truncated to 1024 characters.

## B.5. Forward Operator: Key-wise Context Accumulation with Per-Component Prompt Variables

In the current implementation, the forward pass maintains a shared key–value *context dictionary* that is updated after each component call. At depth $\ell$, a component receives only the fields it declares in `input_fields` (a projection $\Pi_\ell$ of the current context) and returns an output dictionary which is merged into the context by key-wise update.

**Per-component prompt variables.** Each optimizable component contains a prompt text variable (`variable`) that is an optimization target and changes throughout training. In forward execution, the component constructs a lightweight prompt by concatenating `variable` with its current inputs (e.g., "Query", "Evidence", "Hints"), and post-processes the model output (e.g., stripping whitespace or code fences). We therefore omit fixed forward prompt templates in the supplement, since the `variable` content evolves during training and the invariant part is the *interface* (input/output fields) rather than handcrafted wording.

**Connection to backward attribution.** In the backward pass, we replay each component call using the logged trajectory (`input/output`) and instruct the analyzer to treat the component output as the incremental contribution at that step. This aligns forward execution (context accumulation + narrow interfaces) with backward responsibility decomposition (LOCAL vs UPSTREAM) and STOP_GRADIENT pruning.

## B.6. Detailed Context-Window Control

Our current codebase controls context growth primarily through *(a) interface-level projection* and *(b) producer-side hard caps*.

**Interface-level projection.** The system maintains a shared dictionary-like context $h$ during forward execution, but each component is invoked only with the fields listed in its `input_fields`. Thus, $\Pi_\ell(h)$ is implemented as a key-selection step: components do not automatically receive the entire history.

**Producer-side caps.** We bound the main sources of long context at the point of generation: (i) retrieval returns only top-$k$ snippets which are then concatenated into `retrieve_content`; (ii) execution diagnostics (trace/stderr) are truncated to fixed maximum lengths before being consumed by downstream repair. These caps provide a simple and robust window-control mechanism without requiring a global prompt serializer.

**Trajectory logging.** For analysis and backward attribution, we log each component's forward `input` and `output` into a trajectory object (`traj`), enabling post-hoc inspection of what each component saw and produced.

*Table 6.* PromptOptimizer settings.

| Parameter | Setting |
|---|---|
| Optimizer model | openai/gpt-4o-mini |
| Temperature | 0.7 |
| Max new tokens | 512 |

# C. Prompts and System Instructions

This appendix provides the full prompt templates used in TextResNet. We categorize them into three components: (1) The Backward Semantic Projector for failure attribution, (2) The Prompt Optimizer for gradient-based updates, and (3) The auxiliary formatting constraints.

## C.1. Backward Semantic Projector (Attribution)

The *Backward Semantic Projector* acts as a "Root Cause Analysis Agent." It analyzes the execution trace to distinguish between errors caused by the current component (**LOCAL**) and errors propagating from previous steps (**UPSTREAM**).

```
You are an expert failure analyst for a multi-step AI system.
You will analyze a component's trace. Your goal is to pinpoint exactly WHY it failed.
CRITICAL ANALYSIS STEPS:
1. Did the component strictly follow its instructions? If no -> LOCAL fault.
2. Was the input physically insufficient to produce the desired output?  -> UPSTREAM fault.

3. AVOID "HINDSIGHT BIAS": Do not blame the component for not knowing facts that were not
    provided in the context.
4. Distinguish between "Format Error", "Hallucination", and "Missing Info".
Output strictly in the requested format.
```

*Listing 1.* Backward System Prompt (Universal Causal Attribution)

## C.2. Prompt Optimizer

The optimizer uses a meta-prompt to refine the component's system prompt based on batched feedback. This template is designed to prevent overfitting to specific examples by enforcing general rule extraction and negative constraint hardening.

```
You are an expert Prompt Engineer for a Compound AI System.
You will optimize the system prompt for a specific component based on a batch of failure
    feedback from multiple execution traces.

The feedback contains:
1. Context: What the component saw.
2. Local Fix: What the component *should* have done.
3. Upstream Feedback: Ignore this (it belongs to other components).

Your goal: Rewrite the <VARIABLE> prompt to make the component more robust, strictly
    compliant, and immune to the observed failure modes across ALL future inputs.

Component Role:
<ROLE>{variable_desc}</ROLE>

Current Prompt:
<VARIABLE>{variable_short}</VARIABLE>

Batch Feedback:
<BATCH_FEEDBACK>
{variable_context}
</BATCH_FEEDBACK>

ANALYSIS & OPTIMIZATION STRATEGY:
1. **Identify High-Level Patterns**: Look across the batch. Are failures due to format
    violations, logic edge cases, hallucination, or over-truncation?
2. **Avoid Overfitting**: DO NOT mention specific entities, code variables, or exact
    answers from the feedback. Generalize the fix (e.g., instead of "Handle variable x",
```

```
      write "Handle uninitialized variables").
3. **Harden Constraints (Negative Prompting)**: If the model hallucinated or used
   placeholders, add explicit negative constraints (e.g., "NEVER make up facts", "DO NOT
   output markdown fences").
4. **Structural Reinforcement**: Place the most critical rules at the end of the prompt (
   the recency effect for LLMs). Use ALL CAPS for critical constraints.
5. **Maintain Residual Identity**: The component must only output the required increment (
   Delta) as defined in its role. Do not expand its scope.

OUTPUT INSTRUCTIONS:
- You must output the fully rewritten prompt between the {start_tag} and {end_tag} tags.
- The new prompt must be self-contained, clear, and stricter than the original.

{start_tag}{{new_prompt}}{end_tag}
```

*Listing 2.* Prompt Update Template (Batch Optimization)

## C.3. Auxiliary Formatting Constraints

To ensure the stability of the automated pipeline, we employ strict output constraints for the backward pass and a standardized context construction template.

**Optimizer System Message.** This message defines the persona of the meta-optimizer.

```
You are part of an optimization system that improves text (the prompt).
You will receive feedback and context, and use them to improve the prompt.
The feedback may be noisy; identify what is important and correct.
Pay attention to the role description and the context where the prompt is used.
This is very important: You MUST return the improved prompt only between the required tags.
```

**Backward Causal Routing.** This template enforces the parsing format for the feedback router.

```
Output exactly TWO sections, in this exact order:

LOCAL:
(feedback for improving THIS component prompt only; or leave empty)

UPSTREAM:
(STOP_GRADIENT OR feedback for upstream components)

Rules:
- If the failure is mainly caused by THIS component's output (<LM_OUTPUT>), write LOCAL
    and set UPSTREAM to STOP_GRADIENT.
- If the failure is mainly caused by UPSTREAM inputs (<LM_INPUT>), leave LOCAL empty and
    write UPSTREAM feedback.
- If UPSTREAM is STOP_GRADIENT, output ONLY the token STOP_GRADIENT (no punctuation, no
    extra words).
- Do NOT output anything outside these two sections.
```

**Context Construction.** The following template formats the input for the backward operator, presenting the component's trajectory relative to the objective function.

```
You will give feedback to a prompt with the following role:
<ROLE>{variable_desc}</ROLE>

Here is a conversation with a language model:
<LM_SYSTEM_PROMPT>{system_prompt}</LM_SYSTEM_PROMPT>
<LM_INPUT>{lm_input}</LM_INPUT>
<LM_OUTPUT>{lm_output}</LM_OUTPUT>

This conversation is part of a larger system. The <LM_OUTPUT> was later used as {
    response_desc}.
```

```
Treat <LM_OUTPUT> as this component's incremental contribution at this step.
Attribute responsibility by asking: could changing <LM_OUTPUT> (given the same <LM_INPUT>)
    reasonably fix the objective?

<OBJECTIVE_FUNCTION>{objective_feedback}</OBJECTIVE_FUNCTION>

We are interested in giving feedback to the following span of text:
<VARIABLE>{variable_short}</VARIABLE>

Given the above history, route feedback into LOCAL vs UPSTREAM to improve the objective.
```

## D. Theoretical Analysis and Derivations

### D.1. Derivation of Proposition 4.1 (Information Preservation)

**Assumption 4.1 (Independent Routing Policy):** To make the variance analysis tractable, we assume the Semantic Projector's routing decision is statistically independent of the magnitude of the noise realization, depending instead on the structural type of the error. While in practice the projector analyzes the specific textual content, this independence assumption serves as a necessary simplification to model the statistical behavior of the filter over a large distribution of inputs. While specific instances exhibit dependence, over the distribution of all possible inputs and prompts, we model the routing policy's sensitivity as orthogonal to the noise magnitude, treating the filter as a stochastic gate parameterized solely by semantic categories, not signal amplitude.

**Theoretical Justification for exponential decay in standard agentic workflow:** We model the standard agentic workflow as a Markov chain $h_0 \to h_1 \to \cdots \to h_L$, where each transition $M_l$ is a stochastic operator (LLM inference). According to the **Strong Data Processing Inequality (SDPI)** (Polyanskiy & Wu, 2017; Cover, 1999; du Pin Calmon et al., 2017), for any non-trivial stochastic kernel with contraction coefficient $\eta_l < 1$, the mutual information with the input decays exponentially with depth (Schoenholz et al., 2017):

$$I(h_L; h_0) \leq \left( \prod_{l=1}^{L} \eta_l \right) I(h_1; h_0). \tag{8}$$

In lossy rewriting (standard workflows), $\eta_l$ represents the information loss due to summarization or hallucination. In contrast, TextResNet enforces $h_l = h_{l-1} \oplus \Delta_l$. Since the operation $\oplus$ (e.g., concatenation) is structurally invertible regarding $h_{l-1}$, the contraction coefficient is effectively $\eta \approx 1$, preserving the causal path for gradient attribution.

**Proposition 4.1.** *Let $\mathcal{H}$ be the semantic state space. In a deep chain of length $L$ employing additive semantic deltas, the upstream context $h_0$ remains theoretically accessible in the final state $h_L$. Unlike lossy rewriting chains where information decays exponentially with depth, our additive structure ensures that upstream context is available for causal attribution during the backward pass.*

*Derivation.* We analyze the information flow by examining the *invertibility* of the state transition function.

**Case 1: Standard Rewriting (Lossy).** In standard agentic workflows, the state transition is defined as $h_l = \mathcal{M}(h_{l-1}; \theta_l)$, where $\mathcal{M}$ is a generative rewriting process. For the upstream context $h_{l-1}$ to be preserved in $h_l$, the mapping $\mathcal{M}$ must be bijective (invertible). However, LLM summarization or reasoning steps are typically compressive (many-to-one mappings), leading to information loss. Formally, if $\mathcal{M}$ is highly compressive (non-injective), the backward operator cannot uniquely reconstruct the input state $h_{l-1}$ from the gradient signal at $h_l$. In the context of textual optimization, this manifests as Semantic Gradient Vanishing: the optimizer lacks sufficient information to attribute the error to the upstream context $\theta_{l-1}$, effectively breaking the causal link required for effective credit assignment.

Formally, if $\mathcal{M}$ is not invertible, there exists no function $\mathcal{M}^{-1}$ such that $h_{l-1} = \mathcal{M}^{-1}(h_l)$. Thus, the gradient $\nabla_{h_{l-1}} \mathcal{L}$ cannot be accurately computed via the chain rule, as the dependency path is broken.

**Case 2: TEXTRESNET (Preservative).** In our framework, the state transition is defined as an additive operation:

$$h_l = h_{l-1} \oplus \Delta_l, \quad \text{where } \Delta_l = \mathcal{F}(h_{l-1}; \theta_l). \tag{9}$$

In the discrete textual domain, the operator $\oplus$ is instantiated as a structured concatenation or append operation. This operation is structurally invertible with respect to the input history. We can define a trivial projection operator $\Pi_{\text{history}}$ such

that:

$$\Pi_{\text{history}}(h_l) = h_{l-1}. \tag{10}$$

Since $h_{l-1}$ can be perfectly reconstructed from $h_l$ (ignoring practical constraints like context window limits), the information loss is theoretically zero. This ensures that the dependency graph remains fully connected, allowing the Semantic Projector to attribute errors back to $h_0$ regardless of the chain depth $L$. □

### D.2. Derivation of Proposition 4.2 (Bounded Error Propagation)

**Proposition 4.2.** *Let $SNR(g_l)$ be the ratio of actionable signal variance to non-actionable noise variance at depth $l$.* **Assumption:** *We model the error accumulation in the semantic space as a linear additive process with variance $\sigma^2$. Under this simplified model, if there exists a non-zero probability $p$ that a noise component is identified as local-specific and filtered out, the Causal Routing mechanism ensures that the noise variance in $SNR(g_l^{TextResNet})$ is bounded by a constant $\frac{1-p}{p}\sigma^2$ independent of depth $L$. In contrast, without routing, the noise variance grows linearly with $L$.*

**Remark on Linearity Assumption:** We acknowledge that textual error propagation is inherently non-linear. However, consistent with standard theoretical analyses of deep residual networks(He et al., 2016; Hardt & Ma, 2017), we model the error dynamics using a first-order Taylor expansion in the semantic space. This linear approximation allows us to derive tractable bounds on variance reduction, providing theoretical intuition for the stability observed in our experiments.

*Derivation.* We analyze the propagation of error variance under the simplified linear model. Let the total feedback signal at depth $l$ be $G_l = S_l + N_l$, where $S_l$ is the true gradient signal and $N_l$ is the accumulated noise. We focus on the variance of the noise component, $V_k = \text{Var}(N_k)$, where $k = L - l$ represents the distance from the output node.

**Model Assumption.** We assume that at each step, new noise $\delta_k$ is introduced with variance $\sigma^2$, and this noise is uncorrelated with previous noise terms.

**Case 1: Standard Backpropagation (Unbounded Growth).** Without causal routing, the backward operator propagates all incoming feedback. The noise accumulates additively:

$$N_k = N_{k-1} + \delta_k. \tag{11}$$

Based on the independence assumption, the variance at step $k$ is:

$$V_k^{\text{Std}} = V_{k-1} + \sigma^2 = \sum_{i=1}^{k} \sigma^2 = k \cdot \sigma^2. \tag{12}$$

At the input layer ($k = L$), the noise variance is $L\sigma^2$. Since the signal $S_l$ typically does not grow with depth, the Signal-to-Noise Ratio (SNR) decays as $\mathcal{O}(1/L)$, leading to the *Upstream Pollution* problem in deep chains.

**Case 2: TEXTRESNET (Bounded Convergence).** The Semantic Projector implements **Causal Routing**. At each step, with probability $p$, the projector identifies a noise component as purely local to the downstream node and blocks its propagation (Stop Gradient). Let $Z_k \sim \text{Bernoulli}(p)$ be the indicator variable for this routing decision. The noise propagation is modeled as:

$$N_k = (1 - Z_k) \cdot (N_{k-1} + \delta_k). \tag{13}$$

We compute the expected variance $V_k = E[N_k^2]$ (assuming zero mean for simplicity):

$$V_k = E[(1 - Z_k)^2] \cdot E[(N_{k-1} + \delta_k)^2] \tag{14}$$

$$= (1 - p) \cdot (V_{k-1} + \sigma^2). \tag{15}$$

This recurrence relation describes a geometric series. Expanding it yields:

$$V_k = (1-p)\sigma^2 + (1-p)^2\sigma^2 + \cdots + (1-p)^k\sigma^2 \tag{16}$$

$$= \sigma^2 \sum_{i=1}^{k} (1-p)^i. \tag{17}$$

As the chain depth increases ($k \to \infty$), this sum converges to:

$$V_\infty = \sigma^2 \frac{1-p}{1-(1-p)} = \sigma^2 \frac{1-p}{p}. \tag{18}$$

**Conclusion.** The noise variance is bounded by the constant $\frac{1-p}{p}\sigma^2$, which is independent of the total depth $L$. This implies that even in infinitely deep chains, the noise level stabilizes, effectively preventing the divergence of error signals and resolving the *Upstream Pollution* problem. □

## E. Algorithm: TextResNet Optimization

---

**Algorithm 1** TEXTRESNET: Decoupling and Routing Optimization Signals

---

**Require:** Graph $\mathcal{G}$ with $L$ nodes, Dataset $\mathcal{D}$, Reward $R$, Projector $\mathcal{P}$, Scheduler Temp $\tau$.
1: Initialize prompts $\Theta = \{\theta_1, \ldots, \theta_L\}$ and gradient densities $\rho = \{0, \ldots, 0\}$.
2: **for** step $t = 1 \to T$ **do**
3:     Sample batch of inputs $X \sim \mathcal{D}$
4:     // Stage 1: Forward Pass (Additive Semantic Deltas)
5:     **for** $l = 1 \to L$ **do**
6:         Generate semantic delta: $\Delta_l \leftarrow \text{LLM}(h_{l-1}; \theta_l)$
7:         Accumulate state: $h_l \leftarrow h_{l-1} \oplus \Delta_l$ {Identity Highway}
8:     **end for**
9:     Compute global reward and initial gradient: $g_L \leftarrow \text{Critique}(h_L, R)$
10:     // Stage 2: Backward Pass (Semantic Decomposition)
11:     **for** $l = L \to 1$ **do**
12:         Project: $\{g_{\text{local}}, g_{\text{upstream}}\} \leftarrow \mathcal{P}(g_l, \theta_l, h_{l-1})$
13:         // 2a. Handle Local Updates
14:         **if** $g_{\text{local}} \neq \emptyset$ **then**
15:             Accumulate feedback: $\mathcal{K}_l \leftarrow \mathcal{K}_l \cup \{g_{\text{local}}\}$
16:             Update density: $\rho_l \leftarrow \rho_l + 1$
17:         **end if**
18:         // 2b. Causal Routing
19:         **if** $g_{\text{upstream}} \neq \emptyset$ **then**
20:             Route upstream: $g_{l-1} \leftarrow g_{\text{upstream}}$ {Pass-through}
21:         **else**
22:             Stop propagation: $g_{l-1} \leftarrow \emptyset$ {Semantic Stop-Gradient}
23:         **end if**
24:     **end for**
25:     // Stage 3: Density-Aware Optimization Scheduling
26:     **if** $t \pmod{\text{update\_freq}} == 0$ **then**
27:         Sample node $k \sim \text{Boltzmann}(\rho, \tau)$
28:         Update prompt: $\theta_k \leftarrow \text{PromptOptimizer}(\theta_k, \mathcal{K}_k)$
29:         Reset density: $\rho_k \leftarrow 0$
30:         Clear feedback buffer: $\mathcal{K}_k \leftarrow \emptyset$
31:     **end if**
32: **end for**

---

## F. Additional Experimental Analysis

In this section, we provide supplementary experiments to further validate the scalability, robustness, and efficiency of TEXTRESNET. We specifically analyze the impact of chain depth, the choice of the backward optimizer model, the effectiveness of our scheduling strategy, and the token efficiency of the optimization process.

### F.1. Scalability with Chain Depth

A core motivation of TEXTRESNET is to resolve the *Signal Blockage* and *Upstream Pollution* problems that plague standard textual backpropagation in deep workflows. To empirically verify this, we constructed synthetic reasoning chains of varying lengths $L \in \{5, 10, 15, 20\}$ based on the HotpotQA dataset. In order not to break the basic reasoning logic of the system, we add "Identity Node" which are asked to perform lossless re-write of the current output. We compared our method against TextGrad and its summarization variant.

As shown in Figure 7, TEXTRESNET exhibits remarkable stability as the chain depth increases. While the performance of TextGrad decays rapidly due to the accumulation of semantic noise (Attribution Ambiguity) and vanishing gradients, TEXTRESNET maintains a high F1 score even at $L = 20$. This demonstrates that our *Additive Semantic Deltas* and *Identity Highway* effectively preserve the upstream context required for precise credit assignment, regardless of depth. Notably, the summarization baseline (TextGrad + Sum) performs worst in deep chains, suggesting that naive compression further destroys the semantic gradients required for optimization.

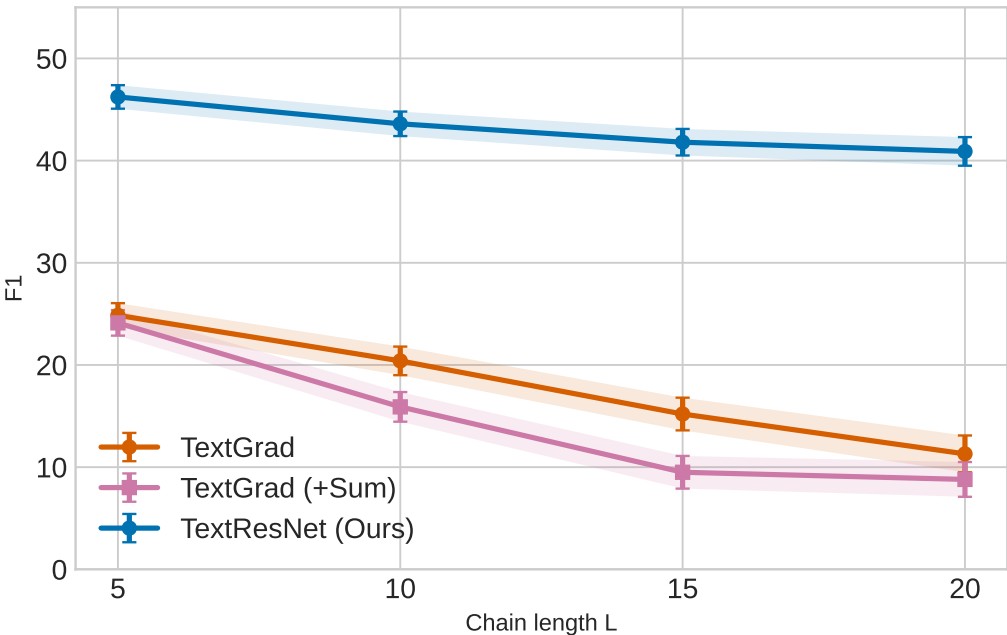

*Figure 7.* **Impact of Chain Depth on Optimization Performance.** We evaluate performance on HotpotQA with artificially added "Identity Node". TEXTRESNET (Blue) maintains performance stability as depth increases, whereas standard TextGrad (Orange) and its summarization variant (Pink) suffer from severe degradation due to semantic entanglement and signal vanishing.

### F.2. Robustness Across Backward Optimizer Models

The *Semantic Projector* in the backward pass serves as a symbolic differentiator. A key question is whether this architecture requires state-of-the-art LLMs to function, or if the structural inductive biases allow for the use of smaller models. We evaluated TEXTRESNET on HotpotQA using three different models for the backward optimizer: Llama-3-8B (Weak), GPT-4o-mini (Mid), and GPT-4o (Strong).

Figure 8 illustrates that TEXTRESNET consistently outperforms the standard TextGrad baseline (dashed line) across all model scales. Even with the relatively weaker Llama-3-8B Instruct, our framework achieves an F1 score significantly higher than the baseline. This indicates that the performance gains are primarily driven by our architectural innovations rather than solely by the reasoning capability of the underlying LLM.

### F.3. Analysis of Scheduling Strategies

To validate the effectiveness of our *Density-Aware Optimization Scheduling*, we compared it against three alternative strategies:

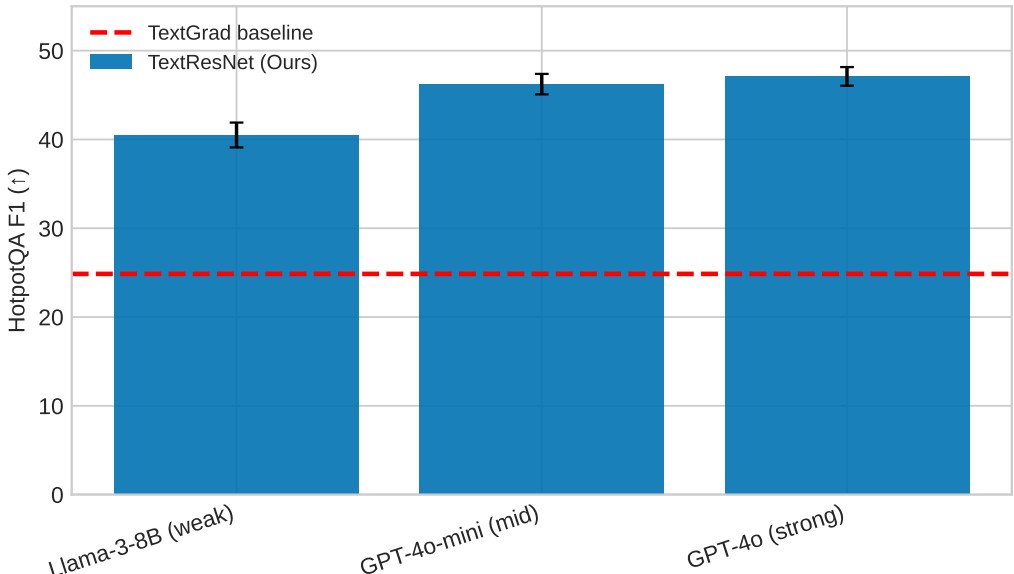

*Figure 8.* **Robustness Across Backward Optimizer Models.** We compare TEXTRESNET performance using different LLMs for the backward Semantic Projector. The red dashed line represents the standard TextGrad baseline. Our method generalizes well even to weaker models (Llama-3-8B Instruct), demonstrating that the structural priors effectively reduce the difficulty of the optimization task.

- **Random:** Randomly selects a component to optimize at each step.

- **Round-Robin:** Iterates through components in a fixed topological order.

- **Greedy:** Selects the component with the highest accumulated error count, but without the Boltzmann exploration mechanism.

The results in Figure 9 show that our Density-Aware approach yields the highest F1 score. While the Greedy strategy performs well, it is prone to getting stuck in local optima by over-optimizing a single noisy component. The Boltzmann sampling in our Density-Aware scheduler balances exploitation (targeting bottlenecks) with exploration, ensuring a more robust global convergence. However, the success of Greedy variant and the Density-Aware scheduler both prove the importance of considering which component to be optimized at each step. The Random and the Round-Robin variants both result in lower score further prove the importance.

### F.4. Token Efficiency and Optimization Cost

Finally, we analyze the computational efficiency of TEXTRESNET. A major limitation of conversational optimization (e.g., TextGrad) is the verbose feedback loop, where every component receives feedback regardless of its contribution to the error.

Figure 10 presents a dual analysis of token consumption. In Figure 10(a), we plot the volume of feedback tokens generated per step. TEXTRESNET (Blue) exhibits a rapid drop in token usage as training progresses. This is a direct result of our *Causal Routing* mechanism: as components converge or when errors are identified as purely local (via STOP_GRADIENT), the Semantic Projector cuts off unnecessary feedback branches. In contrast, TextGrad (Orange) maintains a high volume of entangled feedback throughout the process.

Figure 10(b) summarizes the total cost and performance. TEXTRESNET consumes approximately $3\times$ fewer tokens than TextGrad (21k vs. 63k) while achieving nearly double the performance (46.23 vs. 24.86 F1). This confirms that our method is not only more effective but also significantly more cost-efficient for optimizing Compound AI Systems.

### F.5. Attribution Complexity vs. Chain Depth

The benefit of TEXTRESNET is jointly affected by chain depth and inter-component dependency strength. Depth amplifies attribution ambiguity because more components can inject or inherit errors, but depth alone is not sufficient to predict the

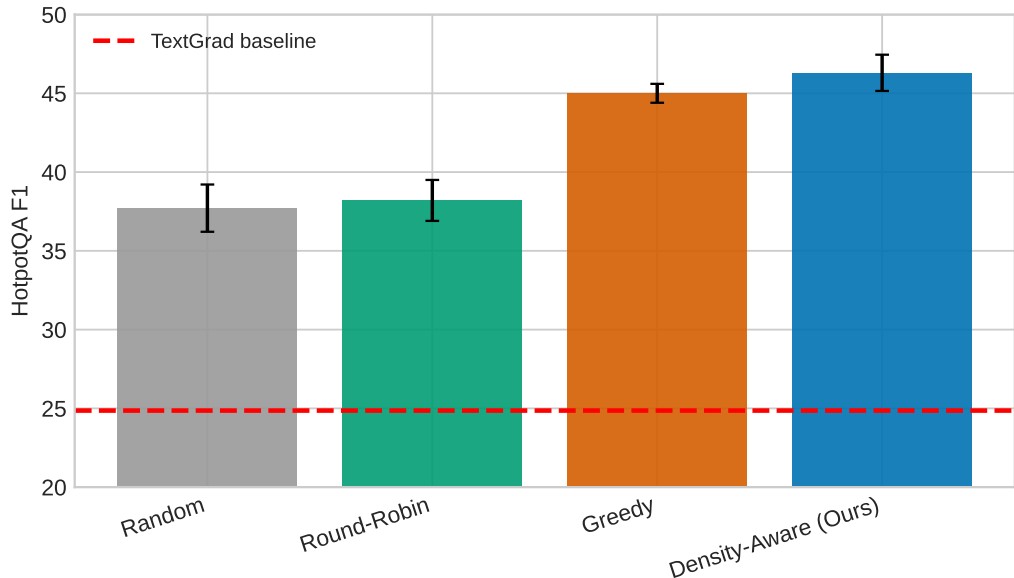

*Figure 9.* **Ablation of Scheduling Strategies.** We compare our Density-Aware Scheduling against Random, Round-Robin, and Greedy baselines on HotpotQA. Our approach (Blue) effectively identifies system bottlenecks while maintaining sufficient exploration, outperforming heuristic baselines.

gain. Across benchmarks, HotpotQA obtains the largest improvement (+21.37 F1) because its retrieval and reasoning stages form a tightly coupled sequential path. PubMedQA and BigCodeBench show moderate gains because their component interactions are shorter or more localized. STaRK obtains the smallest gain (+0.44 MRR) despite using multiple components, because the relation and text scorers operate more independently before final aggregation.

This pattern is consistent with the step-wise ablation in Table 2: the Semantic Projector alone adds +4.54 F1, and combining it with Causal Routing further improves HotpotQA to 37.71 F1 before scheduling is added. These gains isolate the effect of attribution-aware routing from the effect of simply increasing optimization iterations. The chain-depth experiment in Fig. 7 should therefore be interpreted as a stress test for long attribution paths, not as evidence that depth is the only driver.

### F.6. Heterogeneous Model Settings

We test whether TEXTRESNET depends on using the same LLM family in all roles. Table 7 reports HotpotQA results under heterogeneous forward/backward configurations. Performance remains within 1.5 F1 of the default setting, except that a stronger backward model gives a small gain. This supports the model-agnostic design: the architecture constrains how signals are routed, while the backward LLM mainly affects the accuracy of the Semantic Projector.

*Table 7.* Heterogeneous forward/backward model settings on HotpotQA.

| Forward model | Backward model | F1 | Delta |
|---|---|---|---|
| GPT-4o-mini | GPT-4o-mini | 46.23 | – |
| GPT-4o-mini | Claude-3-haiku | 45.10 | -1.13 |
| Mixed GPT-4o-mini/Claude-3-haiku | GPT-4o-mini | 44.80 | -1.43 |
| GPT-4o-mini | GPT-4o | 47.20 | +0.97 |

The mixed-forward setting alternates GPT-4o-mini and Claude-3-haiku across forward components. Its small degradation mainly comes from output-format inconsistencies between model families. The stronger backward model improves F1, consistent with Proposition 4.2: better projector accuracy increases the probability of filtering non-actionable feedback. These experiments test robustness to heterogeneous model roles, rather than feedback aggregation from multiple backward models; multi-model feedback ensembling is left for future work.

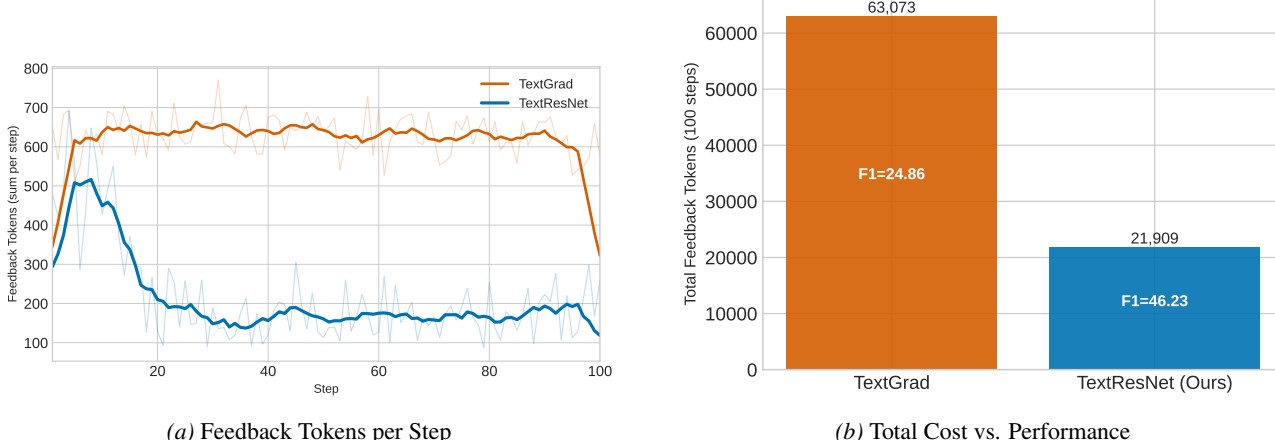

*(a)* Feedback Tokens per Step        *(b)* Total Cost vs. Performance

*Figure 10.* **Token Efficiency Analysis.** (a) The volume of feedback tokens over time. TEXTRESNET naturally reduces feedback volume as the system stabilizes (via `STOP_GRADIENT`), whereas TextGrad continues to propagate noisy feedback. (b) Comparison of total token consumption (100 steps) and final F1 score. Our method achieves superior performance with significantly lower computational overhead.

## F.7. Failure-Mode Annotation Protocol and Examples

To directly evaluate whether TEXTRESNET reduces semantic entanglement, we sampled 50 HotpotQA optimization trajectories and annotated backward feedback messages from TextGrad and TEXTRESNET. Four PhD-level annotators independently labeled each message after a calibration phase on held-out examples. Final labels were assigned by majority vote, followed by a final consistency review. Only one case produced different votes before discussion, indicating that the taxonomy was stable for the sampled outputs.

*Table 8.* Failure-mode distribution in backward feedback on HotpotQA.

| Feedback class | TextGrad | TEXTRESNET | Reduction |
|---|---|---|---|
| Signal Blockage | ∼35% | ∼5% | ∼86% |
| Downstream Over-correction | ∼25% | ∼7% | ∼72% |
| Upstream Pollution | ∼15% | ∼3% | ∼80% |
| Clean feedback | ∼25% | ∼85% | – |

**Classification criteria.** *Signal Blockage* means that feedback reaching an upstream component is too vague to repair the root cause, such as asking a retriever to "improve retrieval of Chevrolet engine types" when the missing keyword is "Big Block." *Downstream Over-correction* means a downstream component is asked to fix an upstream error even though its own behavior was faithful to the input. *Upstream Pollution* means a downstream error is incorrectly sent upstream, causing a previously correct upstream component to change. *Clean feedback* means the responsible component receives specific, actionable instructions.

**Side-by-side example.** For the HotpotQA query "Which engine for medium-duty trucks was also utilized in Chevrolet's intermediate and pony car models?", the ground truth is "The Chevrolet Big Block", while the initial prediction is "Chevrolet small-block V8 engine."

**TextGrad** propagates diluted feedback at each hop:

1. AnswerGenerator: "Review which engine family fits heavier-duty applications."

2. HintGenerator: "Distinguish Chevrolet engine families more specifically."

3. InfoExtractor: "Improve retrieval of Chevrolet engine types."

4. The Retriever never learns to search for "Big Block", leading to Signal Blockage.

**TEXTRESNET** decomposes the same error into local and upstream signals:

1. AnswerGenerator: $g^{\text{local}} = \emptyset$ because the answer is faithful to the hints, while $g^{\text{upstream}}$ states that hints must cite the exact engine family name from evidence.

2. HintGenerator: $g^{\text{local}}$ asks it to cite exact engine family names when present in evidence, while $g^{\text{upstream}}$ asks the Retriever to query "Chevrolet Big Block heavy-duty engine" explicitly.

3. The Retriever receives targeted keywords, resolving the original blockage.

### F.8. Prompt Evolution Patterns and Template Boundary

We analyze prompt snapshots at optimization steps 0, 25, 50, 75, and 100 on HotpotQA using normalized Levenshtein distance between consecutive snapshots. Prompt evolution follows a consistent coarse pattern: early iterations make large edits that improve input-quality checks, middle iterations refine task outputs, and late iterations mainly adjust formatting and boundary conditions.

*Table 9.* Prompt evolution on HotpotQA.

| Steps | Edit distance | Dominant pattern |
|---|---|---|
| 0–25 | 0.416 | Input-quality assessment |
| 25–75 | 0.150 | Output refinement |
| 75–100 | 0.049 | Format and boundary adjustment |

These patterns do not imply that a single generic template can replace optimization. The t-SNE analysis in Fig. 6 shows that optimized feedback clusters are role-specific, such as retrieval-keyword fixes versus reasoning-logic fixes. Different pipeline topologies and evidence sources require different prompt updates. Prompt optimization also has a natural upper bound: it cannot recover information absent from the available evidence or compensate for model capacity limits. This is why stronger backward models can improve the projector in Fig. 8, while the architectural routing still explains the large gap over TextGrad.

### F.9. Cost-Normalized Discussion with OPTIMAS

OPTIMAS (Wu et al., 2025) and TEXTRESNET optimize compound systems through different mechanisms. OPTIMAS learns Local Reward Functions using preference data and RL-based training, whereas TEXTRESNET is a training-free prompt optimizer that routes textual feedback. A strictly equal-cost experiment is therefore structurally difficult: the OPTIMAS budget includes GPU training and preference-data collection, while TEXTRESNET primarily uses LLM API calls.

To make the comparison conservative, we exclude hidden OPTIMAS costs such as preference-data API calls and LoRA initialization, and count only standard generation calls plus base GPU rental. Using a low A100 rental estimate of $2 per GPU-hour, an 8-A100 OPTIMAS run costs $16 per hour in hardware alone. Under the same approximate $20 budget where TEXTRESNET reaches 46.23 F1 on HotpotQA, OPTIMAS can run for only about 1.25 hours, which is less than 20% of its typical 6–8 hour training budget and reaches about 27.00 validation F1 in our truncated run.

We therefore also report gain per cost:

$$\text{Gain per cost} = \frac{\text{Score}_{\text{method}} - \text{Score}_{\text{TextGrad}}}{\text{Total Cost}_{\text{method}}}. \tag{19}$$

*Table 10.* Cost-normalized improvement over TextGrad.

| Method | HotpotQA F1 gain / $10 | BigCodeBench pass-rate gain / $10 |
|---|---|---|
| TEXTRESNET | ~11.1 | ~1.1 |
| OPTIMAS | ~2.0 | ~0.3 |

This analysis suggests that TEXTRESNET provides a roughly 4–6× cost-efficiency advantage under the counted-cost assumptions. The two methods are also complementary: OPTIMAS provides reward-learning infrastructure, while TEXTRESNET provides routing and structural infrastructure. Combining TextResNet-style causal routing with OPTIMAS's learned Local Reward Functions is a promising direction for future work.

# G. Limitations and Future Directions

**Window control is currently simple and heuristic.** Our current context-window control relies on structured state, top-$k$ caps, and truncation. While modern LLMs offer long contexts, these simple mechanisms are not sufficient in the worst case: retrieval evidence may be long-tailed, execution traces can be adversarially verbose, and deep chains can still accumulate many high-priority fields.

**Research directions for stronger window control.** We see several promising improvements:

- **Adaptive budgeting**: learn an instance-conditioned allocator $b_k(x)$ that assigns more budget to salient fields (e.g., via uncertainty, entropy, or learned gates).

- **Causal compression**: compress evidence using attribution-aware objectives (preserve spans that change the decision), rather than generic summarization.

- **State distillation**: distill a compact latent memory (or structured "fact table") that is provably sufficient for downstream components, enabling deeper chains.

- **Retrieval with verification**: replace large evidence blobs with a small set of verified claims + citations, trading length for trustworthiness.

- **Token-level accounting**: enforce explicit token budgets with model-specific tokenizers and overflow-safe fallbacks, rather than character heuristics.

**Broader application scope.** TEXTRESNET is most suitable for modular pipelines with inspectable intermediate states, such as retrieval–reasoning–verification systems, tool-use agents, or code-generation workflows with explicit traces. It is less directly applicable to single-call systems or pipelines where intermediate artifacts are unavailable. The routing mechanism is also complementary to RL-based compound-system optimization: reward-learning methods such as OPTIMAS can learn local objectives, while TEXTRESNET provides a structured mechanism for assigning credit and routing textual feedback across components.

