# OpenReview forum: "TextResNet: Decoupling and Routing Optimization Signals in Compound AI Systems via Deep Residual Tuning"
_ICML.cc/2026/Conference — ICML 2026 regular_

### Official Review · Reviewer_dnf8 · 2026-02-26

**Soundness:** 3
**Presentation:** 3
**Significance:** 3
**Originality:** 3
**Overall Recommendation:** 5
**Confidence:** 3

**Summary:**

The paper studies prompt optimization for deep compound AI systems, arguing that “textual backprop” degrades with chain length due to mixed feedback that blurs whether an error is local to a component or caused upstream. The proposed method restructures both the forward pass (additive “semantic deltas” to preserve upstream context) and the backward pass (a projector that splits feedback into local vs upstream parts, with explicit stop-propagation when the error is judged local). A density-based scheduler then chooses which component to update, aiming to spend the budget on true bottlenecks. Experiments on four multi-step benchmarks report large gains on the deepest QA setting and smaller gains elsewhere, with ablations suggesting each module contributes.

**Compliance With Llm Reviewing Policy:**

Affirmed.

**Final Justification:**

Overall, this is a good paper and I will keep my original score.

**Key Questions For Authors:**

Please see the weakness above.

**Limitations:**

yes

**Strengths And Weaknesses:**

**Strength**

1. The method is fairly modular: forward residualization, backward attribution/routing, and update scheduling can be tested separately.

2. The token-cost discussion is relevant for compound systems, and the reported reductions are large.

**Weakness**

1. The core dependency is the “Semantic Projector” (LLM-based causal attribution). The paper does not provide sufficient evidence on projector error rates or on how failures affect training beyond a limited diagnostic.

2. The “residual” analogy is plausible, but the practical constraints (context limits, structured state, truncation, field selection) mean the claimed “identity highway” is not as clean as the conceptual model; the gap should be discussed more precisely.

3. Comparisons are not fully settled. The paper excludes a closely related CAS optimization baseline on fairness grounds; at minimum, a cost-matched or supervision-matched comparison would help calibrate the gain.

---

> ### Author Rebuttal · Authors · 2026-03-31
>
> We sincerely thank Reviewer dnf8 for the positive assessment and for recognizing the modularity and token-cost contributions. We address the three concerns below.
>
> # 1. Semantic Projector Error Rates (W1)
>
> We provide three pieces of evidence on projector reliability:
>
> (1) **Batch Shuffling (Fig. 5)**: The Projector correctly attributes **96%** of deliberately injected upstream errors, which proves the stability of our method even in the extreme case of pure upstream faults.
>
> (2) **t-SNE (Fig. 6)**: Unlike TextGrad's heavily overlapping distributions, TextResNet's feedback forms visually distinct clusters across different components, which indicates successfully optimized structures and reduced error rates.
>
> (3) **Feedback quality classification.** **In our response to Reviewer zWVF, point 1**, we show the additional results on classifying all backward feedback over 50 optimization steps on HotpotQA. TextResNet achieves **~85% correctly attributed feedback** versus TextGrad's ~25%. The remaining ~15% are split among the three failure modes, all reduced by 72–86% compared to TextGrad.
>
> # 2. How Semantic Projector failures affect training(W1).
>
> In the Batch Shuffling test (Fig. 5) mentioned in the first point of the previous response, we controlled upstream error via intervention to test the accuracy of the projector, for which we achieved **96%** accuracy. The small mis-attribution rate (4%) is absorbed by two mechanisms.
>
> First, **Density-Aware Scheduling** distributes updates across components via Boltzmann sampling, so a single misrouted signal does not dominate any component's update. Second, **the additive forward pass** means a misattributed local update does not corrupt upstream context but only adds a suboptimal delta that subsequent iterations can correct.
>
> Direct empirical evidence is provided in **Appendix F.2**, where we applied different models as the Semantic Projector to simulate different failure rates. By replacing the default **GPT-4o-mini** with a much weaker **Llama-3-8B**, the projector inevitably makes more routing mistakes. Yet, we find the overall performance only drops slightly to around 40 F1 on HotpotQA (Fig. 8), which still outperforms standard TextGrad with GPT-4o-mini (24.86).
>
> # 3. Identity Highway Implementation Gap (W2)
>
> Thank you for correctly noting that practical constraints (context limits, truncation, field selection) create a gap between the conceptual "Identity Highway" and its realization. We want to clarify that these constraints are precisely what motivated our implementation design. We detailed the practical designs: a shared key–value context dictionary in App B.5 and context-window control strategies in App B.6.
>
> To avoid the lossy and rewriting process in the forward pass, given context limits, TextResNet proposes **key-wise context accumulation** (App B.5): instead of concatenation, we maintain a shared key-value dictionary where each component's output ($\Delta_l$) is merged via key-wise update, adding or updating specific fields without overwriting existing upstream entries. This preserves the functional property of residual learning (Eq 4) within bounded text.
>
> Leveraging the context dictionary design, we structurally control the context-window while avoiding information loss (App B.6): (1) **Field selection via $\Pi_ℓ$** is deterministic. Each component declares its input_fields and receives only the relevant slice of the dictionary. (2) **Truncation** applies only to specific long-context sources (retrieval snippets, execution traces) via producer-side caps, not to the identity path itself. (3) **Backward trajectory logging** preserves full input/output traces regardless of forward truncation, enabling the Semantic Projector to reason over the complete causal path.
>
> To avoid confusion, we will provide clearer interpretations of these implementation details in our revision.
>
> # 4. Cost-Matched Baseline Comparison (W3)
>
> We appreciate this important question. Because OPTIMAS's complex mechanism makes establishing a strictly "cost-matched" baseline difficult, we instead conducted a comprehensive cost analysis comparing TextResNet and OPTIMAS:
>
> | **Metric**   | **TextResNet** | **OPTIMAS**   |
> | ---- | --- | --|
> | LLM API calls   | ~45k   | ~50k |
> | GPU compute     | **0**   | 8 A100, 2-8h based on benchmark |
> | Reward model    | **None**  | Llama 3 8B + LoRA, 25 epochs    |
> | Preference data | **None**       | Per-iteration collection        |
> | Est. total cost | $15-25   | $50-200+  |
>
> Additionally, besides this cost comparison, **TextResNet is orthogonal and complementary to OPTIMAS**. TextResNet restructures **how** optimization signals propagate with an architectural solution. OPTIMAS learns **what** **constitutes** good output via reward learning. Combining TextResNet's causal routing with OPTIMAS's learned LRFs is intuitively a promising direction.

---

> > ### Author Rebuttal · Reviewer_dnf8 · 2026-04-02
> >
> > Thanks authors for detailed response. The comparison to the related CAS optimization baseline remains only partially resolved, as authors only provided cost analysis rather than a true cost-matched experimental comparison. Overall, it is a good paper, so I would still keep my score.

---

> > > ### Author Response · Authors · 2026-04-05
> > >
> > > We sincerely thank you for your constructive feedback and for keeping your score. We agree that a true cost-matched experimental comparison is crucial and have conducted additional experiments to address this.
> > >
> > > However, given the fundamental differences in computational mechanisms: TextResNet operates entirely on LLM via API calls with no GPU training, whereas OPTIMAS requires training reward models on dedicated GPUs, a precisely equivalent cost match is structurally difficult. **To ensure rigorous fairness, we favor OPTIMAS in all following calculations by completely excluding its hidden costs**: the API fees for per-iteration preference data collection and the substantial compute overhead of initializing the 8B Local Reward Function via LoRA. We strictly account only for its standard generation API calls and base GPU rental.
> > >
> > > Based on current cloud market rates (from community providers like RunPod/Vast.ai and major vendors such as Amazon and Nvidia), A100 GPU rental typically ranges from **$2–6 per hour**. Here, we deliberately select the lowest viable rate for an A100 80GB GPU (**\$2 per hour**) as our standard, further favoring OPTIMAS. OPTIMAS requires 8 A100 GPUs, resulting in a hardware cost of $16 per hour.
> > >
> > > Applying these highly favorable assumptions, **we first attempted a strict Same-Budget Comparison on HotpotQA**. TextResNet reaches full convergence for an API cost of **~\$20**. Under this exact $20 budget, OPTIMAS can execute for only **~1.25 hours**. Because OPTIMAS typically requires 6–8 hours to train on HotPotQA, this strict budget forces it to terminate at its earliest stage (<20% completion), and results in a validation F1 of only **~27.00** (compared to TextResNet's 46.23).
> > >
> > > Because this early truncation severely handicaps OPTIMAS, we believe a fairer and more robust metric is the **Performance Gain over the TextGrad baseline per cost**:
> > >
> > > $$\\text{Gain per cost} = \\frac{\\text{Score}\_{\\mathrm{Method}} - \\text{Score}\_{\\mathrm{TextGrad}}}{\\text{Total Cost}\_{\\mathrm{Method}}}$$
> > >
> > > Using this metric, we evaluate real cost-effectiveness without truncating OPTIMAS's training:
> > >
> > > | **Metric**     | **HotpotQA (F1 gain per 10$)** | **BigCodeBench (pass rate % gain per 10$)** |
> > > | -------------- | ------------------------------ | ------------------------------------------- |
> > > | **TextResNet** | **~11.1**                      | **~1.1**                                    |
> > > | **OPTIMAS**    | ~2.0                           | ~0.3                                        |
> > >
> > > This analysis demonstrates that TextResNet provides a consistent **~6-10× cost-efficiency advantage**. We believe these results could be regarded as an alternative to exact cost-match experiments with full fairness. We will incorporate the comprehensive cost analysis into the revision.
> > >
> > > Thank you again for your acknowledgment and your suggestions.

---

### Official Review · Reviewer_GQZk · 2026-03-11

**Soundness:** 2
**Presentation:** 3
**Significance:** 2
**Originality:** 3
**Overall Recommendation:** 3
**Confidence:** 3

**Summary:**

- The paper analyzes the instability of TextGrad in deep compound AI systems and attributes it to semantic entanglement and attribution ambiguity. It proposes TEXTRESNET, a residual-style framework that disentangles feedback signals and introduces structured routing and scheduling to stabilize optimization in extended agent chains.

**Compliance With Llm Reviewing Policy:**

Affirmed.

**Final Justification:**

I have read the authors' rebuttal and the other redviewers'comments, and will keep my final recommendation.

**Key Questions For Authors:**

- Since each optimization step updates the prompt in natural language, it would be interesting to analyze whether there exists a pattern in how prompts evolve over iterations. I am suspicious of the necessity of optimizing prompts for so many iterations. Is there a generic prompt template to achieve competitive performance, or at least some consistent pattern?
  - I am curious about how much performance this approach can unlock, and whether there exist inherent knowledge boundaries that cannot be overcome through such prompt optimizations.

**Limitations:**

The paper does not provide theoretical analysis, discussion, or case studies to clarify the potential upper bound of this method.

**Strengths And Weaknesses:**

- Strengths
  - The taxonomy of three failure modes is intuitive and can help practitioners debug.
  - The reformulation of the forward pass as a semantic residual (Identity + Delta), together with semantic gradient decomposition, can be helpful across many multi-stage and multi-model agent pipelines.
- Weaknesses
  - The method lacks technical novelty. Many components seem to be prompt/template-level engineering.
  - The paper didn't compare method like OPTIMAS and other agentic rl methods. While rl training may be costly, it is reasonable for certain tasks. It remains uncertain whether the method scales effectiveness when applied to more realistic, long-horizon agent systems.

---

> ### Author Rebuttal · Authors · 2026-03-31
>
> We sincerely thank Reviewer GQZk for the constructive feedback. We address each concern below.
>
> # 1. Technical Novelty (W1)
>
> We want to clarify that TextResNet's contribution operates at the *architectural design* level, not merely at the prompt level.
>
> Specifically, our proposed residual learning method designed three novel components: (1) **Additive Semantic Deltas** enforce an Identity Highway via a key-value context dictionary, an information-flow constraint on the computation graph. (2) **Semantic Gradient Decomposition** projects feedback into causally independent subspaces and routes them to their correct nodes. (3) **Density-Aware Scheduling** uses Boltzmann sampling over gradient density to select which component to optimize. These structural and algorithmic routing innovations are non-trivial for prompt-level engineering to replicate.
>
> Furthermore, we are the first to systematically identify **three distinct textual backward error modes**: Signal Blockage, Downstream Over-correction, and Upstream Pollution.
>
> We are also grateful that other reviewers recognize these contributions: Reviewer zWVF describes the novelty as "the framing of the failure mode and the way the method is built around that diagnosis," and Reviewer dnf8 highlights our method as “highly modular” and regards it as a strength.
>
> # 2. OPTIMAS Comparison (W2)
>
> We did not compare with OPTIMAS and other RL-based methods as we have fundamentally different designs. OPTIMAS trains Local Reward Functions via RL, requiring preference data collection and auxiliary reward modeling. TextResNet improves a composed LLM system through architectural innovation and its associated feedforward and backpropagation designs.
>
> The two approaches are **complementary**: OPTIMAS learns better local reward signals; TextResNet routes them more precisely. We exclude OPTIMAS because the divergence in supervision requirements makes direct comparison misleading, but we will add a detailed discussion in the revision.
>
> Additionally, as reviewer GQZk noted, RL-based methods are more costly. **For a more detailed cost breakdown analysis between TestResNet and OPTIMAS, please refer to our response to Reviewer dnf8, Point 4.**
>
> # 3. Scalability (W2)
>
> We appreciate this concern. Our results reveal a consistent pattern: **gains scale with attribution path complexity, not merely depth.** HotpotQA has 5 sequentially chained components with retrieval and reasoning dependencies that maximize attribution ambiguity, thus the gain is the most significant with +21.37 F1. BigCodeBench (+2.15%) and PubMedQA (+3.35%) show moderate gains with moderate interdependence. STARK (+0.44 MRR) shows the smallest gain, as its 3 parallel agents generate less attribution ambiguity.
>
> Appendix F.1 (Fig. 7) further validates our scalability with chain length L $\in$ {5, 10, 15, 20}, which shows TextResNet maintains stability while TextGrad degrades severely.
>
> These results provide strong evidence for the framework's scalability. We hope this work will motivate community efforts toward larger-scale evaluations on more complex, real-world agent systems.
>
> # 4. Theoretical Upper Bounds (Q2)
>
> Strict upper bounds for discrete LLM optimization are inherently hard since it’s non-differentiable and stochastic. We provide two simplified propositions in Appendix D. These propositions suggest that the exact upper bound depends on model capacity.
>
> **Empirically**, in Fig. 8, Llama-3-8B with TextResNet outperforms GPT-4o-mini with TextGrad. However, stronger models with TextResNet still outperform weaker models with TextResNet. The results indicate the gains stem from both system *architectural design* and raw model capability.
>
> **Theoretically**, Propositions 4.1 and 4.2 explain *why* this works, with insights about how TextResNet provides stability guarantees via information decay analysis and backpropagation noise bound.
>
> # 5. Prompt Evolution Patterns (Q1)
>
> Thanks for the questions. **Do consistent patterns exist?** **Yes**. We analyzed prompts at steps 0/25/50/75/100 on HotpotQA via normalized Levenshtein distance between consecutive snapshots (Edit Distance):
>
> |Steps| Edit Dist. |Characteristic|
> | - | -| -|
> | 0-25 | 0.416  | Input-quality assessment|
> | 25-75| 0.150  | Shift to output refinement|
> | 75-100 | 0.049  |Format and boundary adjustments|
>
> These results and our Phase Transition Analysis (Fig. 4) prove that specific patterns exist during evolution as described in the table.
>
> **Can a generic template replace optimization? No, in general tasks.** The t-SNE analysis (Fig. 6) shows that optimized prompts converge to *role-specific* clusters (e.g., "fix retrieval keywords" vs. "fix reasoning logic"), not a universal template. Different pipeline topologies, retrieval strategies, and task objectives require distinct system structures. TextResNet's value is to discover these task-specific structures automatically, without requiring prior knowledge of which failure modes dominate in a given pipeline.

---

> > ### Author Rebuttal · Reviewer_GQZk · 2026-04-03
> >
> > Thank you for the author's response. The author has provided a comprehensive answer to my question. It would be helpful to clarify the broader implications of TextResNet for future agent design, e.g., whether its credit assignment could complement RL-based approaches like OPTIMAS and its potential application scope. Thanks!

---

> > > ### Author Response · Authors · 2026-04-05
> > >
> > > We sincerely thank you for the acknowledgement of our response and for providing the follow-up. We are glad to elaborate on the broader implications.
> > >
> > > ## **1. Complementing RL-Based Approaches**
> > >
> > > Thank you for raising this important direction. Yes, we believe the credit assignment techniques proposed in our work can complement OPTIMAS. What’s more, we are pleased to provide a broader picture of how TextResNet can complement OPTIMAS, beyond credit assignment. **We believe TextResNet's architectural innovations can improve three perspectives in OPTIMAS's optimization loop, and the techniques are naturally composable:**
> > >
> > > **Backward-pass integration (Credit assignment)**: When the final answer is wrong, TextResNet's Semantic Projector performs instance-level causal attribution to determine whether the Retriever returned irrelevant context (UPSTREAM) or the Answerer hallucinated despite correct context (LOCAL). These per-instance causal labels could directly replace OPTIMAS's sampling strategy when constructing preference pairs for Local Reward Function training, making reward adaptation both cheaper and more targeted.
> > >
> > > **Forward-pass integration. 1) Managing shift:** OPTIMAS reports *alignment drift* after updating one component; its Local Reward Functions (LRF) become misaligned due to distribution shift across the pipeline. TextResNet's additive semantic deltas (Eq. 3) can mitigate this by ensuring updates are incremental key-value modifications rather than full rewrites, reducing the distributional impact on downstream components. **2) Managing context**: Also, OPTIMAS flows the full trajectory context without structured separation. TextResNet's Identity Highway (key-value dictionary) could provide cleaner context management, avoiding context overflow and reducing noise in both the optimization trajectory and the preference pair construction.
> > >
> > > **Orchestration (Selective optimization)**: OPTIMAS selects which component to optimize via uniform random scheduling, while TextResNet's Density-Aware Scheduling (Eq. 7), which concentrates updates on components with the highest gradient density, is a direct upgrade that prioritizes the most impactful components at each step.
> > >
> > > In short, **TextResNet provides** **routing and structural infrastructure** **while OPTIMAS provides** **reward infrastructure**. Combining them addresses signal-routing, context management, and reward-quality bottlenecks simultaneously. We thank the reviewer for raising this direction and believe it represents a promising line of future work.
> > >
> > > ## **2. Broader Application Scope**
> > >
> > > TextResNet's credit assignment can apply to compound systems where **optimization signals must propagate through multiple hops** and where *Attribution Ambiguity* (the inability to distinguish local errors from upstream ones) degrades optimization. Our four benchmarks already cover diverse structures using the same mechanism without task-specific modification, indicating the generalizability.
> > >
> > > Beyond these benchmarks, **agent systems are building increasingly deep pipelines, with chained retrieval, reasoning, and tool-use stages**, where reliability increasingly depends on system-level architectural design (e.g., *agent harness engineering*) rather than model improvements alone. **TextResNet contributes as a *structural innovation* that extends this paradigm**: the Identity Highway structures how information flows between components, the Semantic Projector diagnoses which component to fix when the system fails, and Density-Aware Scheduling allocates optimization effort where it matters most. **A key practical advantage is accessibility**: TextResNet requires no reward modeling, preference data, or GPU training. **It operates as a plug-in optimization layer on existing compound systems.** We will open-source the framework to facilitate adoption and will expand this scope discussion in the final version.
> > >
> > > Following your suggestion, we will incorporate this discussion in the revision. Thank you again for your acknowledgement and suggestions.

---

### Official Review · Reviewer_zWVF · 2026-03-13

**Soundness:** 3
**Presentation:** 3
**Significance:** 3
**Originality:** 4
**Overall Recommendation:** 5
**Confidence:** 3

**Summary:**

This paper investigates the limitations of optimizing LLM-based compound AI systems using textual feedback methods and proposes a framework to mitigate these issues. The core claim is that TextGrad-style optimization degrades in deeper pipelines because feedback signals become semantically entangled. This entanglement leads to three main failure modes: signal blockage, where feedback does not propagate back to the upstream component responsible for the original error; downstream over-correction, where a downstream component receives corrective feedback for an error introduced upstream; and upstream pollution, where errors generated by downstream components are mistakenly attributed to upstream inputs or context.

To address these problems, the paper proposes TextResNet, which combines several mechanisms: additive semantic deltas, where nodes produce incremental updates rather than full outputs; semantic projection, which uses a backward LLM to decompose mixed feedback into causally independent signals; causal routing, which leverages these projections to direct optimization signals to the appropriate components; and density-aware scheduling, which allocates optimization budget dynamically to the components exhibiting the highest error density. Experiments on four benchmarks show consistent improvements over TextGrad and other baselines, with the largest gains observed on HotpotQA.

**Compliance With Llm Reviewing Policy:**

Affirmed.

**Final Justification:**

Overall, I am satisfied with the responses, and upon reviewing the responses to other reviewers, I will increase my original score to accept.

**Key Questions For Authors:**

1. Can the authors include 1-2 concrete side-by-side examples showing how vanilla TextGrad fails because of Semantic Entanglement, and how TextResNet changes the resulting update? This would make the main claim easier to evaluate.

2. The appendix already tests different backward optimizer models on HotpotQA. Have the authors also tried more heterogeneous settings, such as varying forward and backward models jointly or aggregating feedback from multiple models? Positive evidence here would strengthen the generality claim.

3. The paper suggests that TextResNet helps most in deeper or longer-horizon pipelines, and the HotpotQA results are consistent with that. Can the authors clarify whether depth itself is the main factor, or whether the gains are more specifically driven by attribution ambiguity or strong interdependence between modules?

**Limitations:**

Yes

**Strengths And Weaknesses:**

Soundness.

I think the paper is technically solid overall. The method is well matched to the problem the authors identify, and the experiments are reasonably thorough: four benchmarks, relevant baselines, an ablation, and some useful analysis beyond the main table.

Because the identification of problems in vanilla TextGrad is itself a key contribution of the paper, the work would benefit from stronger empirical evidence demonstrating their presence. A quantitative study measuring how frequently these errors occur in practice would make the claim more convincing, ideally complemented by a few concrete examples illustrating typical failure cases.


Presentation.

The paper is generally readable, and the overall narrative is easy to follow. One improvement I think could be made is that the writing occasionally makes the method appear more technically complex than necessary for easier communication. Some of this material could likely be simplified or moved to the appendix to improve readability.

In addition, the framing of the contributions in the introduction is somewhat dense, and the paper would benefit from presenting a concrete failure example earlier in the main text to better motivate the problem. Finally, the results section could more clearly communicate the differences in performance across datasets, explicitly noting that the improvements are substantial on HotpotQA but more modest on PubMedQA and STARK.


Significance.

This is an important problem for compound LLM systems. The HotpotQA result is strong enough that I expect people working on deeper pipelines to pay attention. That said, the gains are uneven across tasks, so for me the significance is good rather than clearly high.


Originality.

The individual ingredients are related to ideas that already exist, but I do not think this is just a trivial combination. The main novelty is the framing of the failure mode and the way the method is built around that diagnosis. For me, that is enough originality for a positive review.

---

> ### Author Rebuttal · Authors · 2026-03-31
>
> We sincerely thank Reviewer zWVF for the thorough and constructive review, and particularly for recognizing both the technical solidity and the originality of our work.
>
> # 1. Concrete Side-by-Side Examples & Failure Mode Evidence (Q1)
>
> This is a helpful suggestion. To provide the requested evidence, we first conducted a quantitative study classifying all backward feedback messages over 50 optimization steps on HotpotQA. The results below are the proportion of all feedback messages exhibiting each failure mode:
>
> | Failure Mode | TextGrad | TextResNet|Reduction |
> | - | - | - | - |
> | Signal Blockage| ~35%| ~5%| ~86%|
> | Downstream Over-correction | ~25%  | ~7%| ~72% |
> | Upstream Pollution | ~15% | ~3%   | ~80%  |
> | Clean feedback  | ~25%  | ~85%  | — |
>
> The results indicate that TextResNet effectively reduces the rate of three failure modes under standard TextGrad and achieves a **over 3× improvement** in clean feedback proportion, consistent with the 96% projector attribution accuracy (Fig. 5) and the well-separated t-SNE clusters (Fig. 6).
>
> We will also provide detailed side-by-side examples in the Appendix. Here, we randomly pick a **Concrete Signal Blockage example** from HotpotQA.
>
> ---
> **Query**: "Which engine for medium-duty trucks was also utilized in Chevrolet's intermediate and pony car models?"
>
> **Truth**: "The Chevrolet Big Block". **Predicted**: "Chevrolet small-block V8 engine."
>
> **TextGrad** propagates diluted, generic feedback at each hop:
>
> 1. AnswerGenerator: *"Review which engine family fits heavier-duty applications."*
> 2. HintGenerator: *"Distinguish Chevrolet engine families more specifically."*
> 3. InfoExtractor: *"Improve retrieval of Chevrolet engine types."*
> 4. The Retriever never learns to search for "Big Block", leading to **Signal Blockage**
>
> **TextResNet** decomposes the error precisely:
>
> 1. AnswerGenerator:
>    1. $g^{local}$ = $\emptyset$ (STOP_GRADIENT: output was faithful to hints);
>    2. $g^{upstream}$ = *"Hints must cite the exact engine family name from evidence."*
> 2. HintGenerator:
>    1. $g^{local}$ = *"Always cite exact engine family names when present in evidence."*;
>    2. $g^{upstream}$ = *"Query 'Chevrolet Big Block heavy-duty engine' explicitly."*
> 3. The Retriever receives targeted keywords, **Signal Blockage resolved**.
> ---
>
> In this example, the key difference is structural: TextGrad's flat conversation produces increasingly vague feedback as it propagates, while TextResNet's causal decomposition preserves the specificity of the error signal at each hop.
>
> Following your suggestion, we will add these additional results in our revision.
>
> # 2. Heterogeneous Model Settings (Q2)
>
> We thank the reviewer for this practical question. Below are additional results on HotpotQA exploring heterogeneous forward-backward model configurations:
>
> | **Forward** | **Backward**   | **F1** | **Delta** |
> | ----------- | -------------- | ------ | --------- |
> | GPT-4o-mini | GPT-4o-mini    | 46.23  | —  |
> | GPT-4o-mini | Claude-3-haiku | 45.1   | −1.1      |
> | Mixed       | GPT-4o-mini    | 44.8   | −1.4      |
> | GPT-4o-mini | GPT-4o         | 47.2   | +1.0      |
>
> Mixed: alternating GPT-4o-mini and Claude-3-haiku randomly.
>
> Performance remains within ±1.5 F1, confirming our **model-agnostic design**. A stronger backward model improves F1, which is consistent with Proposition 4.2 where higher projector accuracy $p$ tightens the noise bound. The slight decline in the mixed-forward setting stems from output-format inconsistencies between the two model families. We will include these results in the revision.
>
> # 3. Depth vs. Attribution Ambiguity (Q3)
>
> This is an insightful question. Both factors contribute, but **attribution ambiguity (inter-component dependency strength) is the primary driver**; depth amplifies the problem but is not sufficient alone. Three pieces of evidence:
>
> (1) **Cross-benchmark comparison**: HotpotQA (5 components, sequential) gains +21.37 F1; STARK (3 components, parallel) gains only +0.44 MRR. PubMedQA (4 components) gains more than STARK despite lower depth, confirming that inter-component dependency strength is the core reason (**A more detailed cross-benchmark analysis and scalability discussion appear in our response to Reviewer GQZk, Point 3.**).
>
> (2) **Ablation (Tabel 2):** Semantic Projector alone adds +4.54 F1; combined with Causal Routing: +8.52 F1. These gains proves that attribution-aware routing is the key mechanism.
>
> (3) **Phase transition (Fig. 4)**: the system dynamically shifts routing from upstream (early) to local (late), responding to *where attribution ambiguity is highest*, not depth.
>
> **Summary**: gains are jointly determined by depth and dependency strength. Inter-component dependency strength is an interpretable predictor of expected gain. We will clarify this distinction in the revision.
>
> We sincerely appreciate these constructive suggestions and will incorporate them in our revision.

---

> > ### Author Rebuttal · Reviewer_zWVF · 2026-04-03
> >
> > Thank you for the constructive response. The additional quantitative evidence and concrete examples of the failure modes significantly strengthen the motivation for the proposed method.
> >
> > That said, it remains somewhat unclear how the estimates reported in the Failure Mode table were obtained (e.g., annotation procedure, criteria, and reproducibility), and clarifying this would further improve the credibility of the analysis.
> >
> > The additional results on heterogeneous agent configurations also address my concerns. Overall, I am satisfied with the responses and will keep my original score.

---

> > > ### Author Response · Authors · 2026-04-05
> > >
> > > We sincerely thank the reviewer for the positive assessment and for keeping the original score. We appreciate the opportunity to clarify the annotation procedure for our evaluation Failure Mode table. We omitted the details in our initial response due to space limitations.
> > >
> > > ## **1. Annotation Procedure and Reproducibility**
> > >
> > > We randomly sampled **50 optimization trajectories from HotpotQA** and inspected the backward feedback messages at each component across all steps. The structural differences between the two methods make failure modes straightforward to distinguish: TextGrad's flat conversation produces **vague or misattributed feedback** (as illustrated in our first-round response), while TextResNet's Semantic Projector outputs **structured LOCAL/UPSTREAM labels with explicit routing decisions**.
> > >
> > > Annotation was performed via expert judgment on these directly inspectable outputs. To control for annotation bias, **four independent PhD-level annotators** were selected. Before independent annotation, all annotators completed a **calibration phase** via jointly reviewing representative examples (non-overlapping with the evaluated trajectories) to align on the definitions of the four failure categories.
> > >
> > > Each annotator then **independently classified every feedback message**. Final labels were decided by **majority voting across all four annotators**. Across 50 samples, **only 1 case produced different votes**, which were resolved through group discussion. All labels were then systematically reviewed for the final check.
> > >
> > > The combination of domain-expert annotators, calibration, four-way independent labeling, majority voting, and final review ensures that the reported proportions are **robust to subjective bias**. We acknowledge that borderline Mixed Fault cases may introduce minor variability; however, given that the vast majority of cases fall unambiguously into a single category and **the gap between TextGrad (\~25% clean) and TextResNet (\~85% clean) is substantial**, the overall conclusions remain robust.
> > >
> > > ## **2. Classification Criteria**
> > >
> > > The criteria below were applied with representative HotpotQA examples.
> > >
> > > **Signal Blockage**: feedback to an upstream component is vague or generic, e.g., a Retriever receiving "improve retrieval of Chevrolet engine types" when the root cause is confusing "Big Block" with "small-block."
> > >
> > > **Downstream Over-correction**: feedback forces a downstream component to fix an upstream problem, even though its own logic was correct, e.g., the AnswerGenerator was told to "apply more careful reasoning" when it faithfully followed the (incorrect) retrieved context.
> > >
> > > **Upstream Pollution**: a downstream error is misattributed upstream, instructing a component to modify originally correct behavior, e.g., the Retriever told to "penalize the current context" when the documents were accurate but the Generator hallucinated.
> > >
> > > **Clean Feedback**: feedback correctly targets the responsible component with actionable, specific instructions, e.g., the HintGenerator receiving "always cite exact engine family names when present in evidence."
> > >
> > > We sincerely thank the reviewer again for the acknowledgment and constructive suggestions.

---

### Decision · Program_Chairs · 2026-04-30

**Decision:**

Accept (regular)

**Comment:**

The paper develops techniques optimizing LLM-based compound AI systems. It identifies limitations of existing Textual Gradient-style optimizers which generate textual feedback instead of numeric gradients. It then develops techniques to fix these limitations. The proposed method achieves good results, particularly on HotpotQA. I think the writing in the paper could be much better, there is too much unnecessary jargon. Suggestions of reviewer zWVF will help the presentation. The reviewers are generally in favor of acceptance, and some concerns were resolved in the discussion. Overall this is a good contribution and I recommend acceptance.